# SparseD: Sparse Attention for Diffusion Language Models

**Zeqing Wang**[1], **Gongfan Fang**[1], **Xinyin Ma**[1], **Xingyi Yang**[2*], **Xinchao Wang**[1*]

[1]National University of Singapore, [2]The Hong Kong Polytechnic University

`zeqing.wang@u.nus.edu, xingyi.yang@polyu.edu.hk,`
`xinchao@nus.edu.sg`

## Abstract

While diffusion language models (DLMs) offer a promising alternative to autoregressive models (ARs), existing open-source DLMs suffer from high inference latency. This bottleneck is mainly due to the attention's quadratic complexity with respect to context length in computing all query–key pairs. Intuitively, to reduce this complexity, a natural strategy is to restrict attention to sparse patterns that retain only the most relevant connections. Such approaches are well-established in ARs, where attention follows fixed and clearly defined sparse patterns. However, in DLMs, we observe distinct sparsity behaviors: (1) attention patterns vary across heads, (2) attention patterns in each head remain highly similar across denoising steps, and (3) early denoising steps are critical for generation. These findings render sparse attention methods designed for ARs largely incompatible with DLMs, as they fail to capture head-specific structures and risk degrading generation when applied in early denoising steps. To address these challenges, we propose **SparseD**, a novel sparse attention method for DLMs. Leveraging the observations, SparseD only requires pre-computing head-specific sparse patterns one time, and reuses them across all steps. This prevents recomputing sparse patterns at each denoising step. Meanwhile, SparseD uses full attention in the early steps, then switches to sparse attention later to maintain generation quality. Together, these establish SparseD as a practical and efficient solution for deploying DLMs in long-context applications. Experimental results demonstrate that SparseD achieves lossless acceleration, delivering up to $1.50\times$ speedup over FlashAttention at a 64k context length with 1,024 denoising steps. Code is available at `https://github.com/INV-WZQ/SparseD`.

## 1 Introduction

Recently, diffusion language models (DLMs) have achieved significant progress in the area of natural language processing (Nie et al., 2025; Ye et al., 2025). Unlike traditional autoregressive models (ARs) (Touvron et al., 2023; Yang et al., 2025), which generate tokens sequentially from left to right, DLMs generate the entire context in parallel. Leveraging this capability, DLMs achieve strong performance in language generation and represent a promising alternative to ARs.

Despite the advantages of parallel decoding, DLMs suffer from high-generation latency (Ma et al., 2025; Wu et al., 2025). This bottleneck arises mainly from the bidirectional attention mechanism (Vaswani et al., 2017), which is central to DLMs. This mechanism computes attention over all query–key token pairs simultaneously, including both prefill (prompt) and all generation tokens. As context length increases, the complexity of this mechanism grows quadratically, leading to high latency in generation and limiting the efficiency of DLMs in real-world applications.

To reduce this complexity, sparse attention methods (Xiao et al., 2023; Lai et al., 2025) have emerged as an effective solution. These methods lower the cost of standard attention by restricting computations to sparse patterns that include only a subset of important query–key pairs, i.e., important attention scores. Such approaches have been widely adopted in ARs, as attention in ARs exhibits

---

*Corresponding Author.

prominent and fixed sparse patterns (Xiao et al., 2023). Therefore, applying such methods to DLMs first requires verifying whether sparse patterns also exist in their attention mechanisms.

In this paper, we investigate attention patterns in DLMs and find that they also exhibit clear sparse patterns, making sparse attention feasible in theory. However, we identify three unique observations in DLMs: (1) attention patterns vary significantly across attention heads, showing head-specific patterns, (2) attention patterns within each head remain highly consistent across denoising steps, (3) early diffusion steps are critical for generation, rendering sparse attention unsuitable at this stage. These unique observations make sparse attention methods designed for ARs largely incompatible with DLMs. Widely used fixed patterns in ARs, such as the sliding-window scheme (Jiang et al., 2023) and sink attention (Xiao et al., 2023), fail to capture head-specific patterns of DLMs. Moreover, applying sparse attention to DLMs in the early steps leads to degradation in generation quality.

To tackle these problems, we introduce **SparseD**, a novel sparse attention approach tailored for DLMs. Its core principle is to efficiently handle the unique attention patterns of DLMs without degrading generation quality. To achieve this goal, the three empirical observations outlined above serve as the foundation. Specifically, SparseD pre-computes and selects important query–key pairs for each head once to construct head-specific sparse patterns. These sparse patterns are then reused for sparse attention across denoising steps without the need to recompute. To enable hardware-friendly acceleration, we select important pairs as block-wise query–key pairs (Dao, 2023) rather than individual pairs. Besides, SparseD applies full attention in the early steps to prevent significant degradation in generation quality. These designs enable SparseD to capture head-specific dynamics without incurring significant latency in recomputing sparse patterns at every denoising step, while also preventing the generation degradation caused by sparse attention in the early steps.

To further preserve accuracy, we adopt an isolated selection strategy in computing sparse patterns. Specifically, we observe that attention scores for generation tokens are relatively low during the early steps but gradually increase in later steps. Since SparseD computes sparse patterns in the early steps and reuses them in subsequent steps, this will cause the selection to concentrate primarily on prefill tokens with high attention scores. To address this issue, we separately select important scores for prefill and generation tokens, ensuring that both receive sufficient attention in selection.

Together, these establish SparseD as a practical and efficient solution for deploying DLMs, particularly in long-context applications. Experiments on recent DLMs, including Dream-7B-Instruct (Ye et al., 2025) and LLaDA-1.5 (Zhu et al., 2025), demonstrate that SparseD greatly preserves the original accuracy with negligible loss while achieving up to $1.50\times$ speedup over FlashAttention (Dao, 2023) at a 64k context length with 1,024 diffusion steps.

In summary, our main contributions are as follows:

- We identify three key observations in DLMs: (1) attention patterns vary across heads, (2) attention patterns remain highly consistent across denoising steps, and (3) early diffusion steps are crucial for effective language generation.
- We propose an effective and efficient sparse attention method that skips sparse attention during early denoising steps and reuses sparse patterns in the subsequent steps.
- Extensive experiments show that SparseD greatly maintains accuracy on the evaluated benchmarks while achieving up to $1.50\times$ speedup at a 64k context length with 1,024 steps.

## 2 RELATED WORKS

**Diffusion Language Models (DLMs)**   Diffusion models (Ho et al., 2020; Rombach et al., 2022) have emerged as a powerful paradigm in generative modeling, framing data generation as the inversion of a forward-noise process. They have achieved remarkable success in continuous domains such as images (Peebles & Xie, 2023) and videos (Kong et al., 2024). More recently, diffusion models have also advanced the natural language processing area. DLMs (Ye et al., 2025; Nie et al., 2025) extend diffusion to discrete sequences by redefining noise injection and denoising(Ou et al., 2025; Zheng et al., 2023). Unlike conventional autoregressive models (ARs) (Touvron et al., 2023; Yang et al., 2025) that generate tokens sequentially, DLMs denoise all tokens jointly in a bidirectional manner. This parallel, bidirectional generation enables DLMs to achieve strong performance in both language understanding and generation, establishing them as a promising alternative to ARs.

**Sparse Attention** Despite their success, DLMs suffer from high inference latency, which remains a major bottleneck. This issue is primarily due to the quadratic complexity of the core attention mechanism (Vaswani et al., 2017). This challenge has been extensively studied in traditional ARs. To address it, sparse attention has emerged as a promising and mature solution. In ARs, many methods restrict attention computation to fixed patterns, such as sink attention (Xiao et al., 2023) and the sliding-window approach (Jiang et al., 2023). Some approaches, such as those in BigBird (Zaheer et al., 2020) and Longformer (Beltagy et al., 2020), utilize global and sliding-window attention schemes. Other approaches (Zhang et al., 2025; Lai et al., 2025) identify distinct fixed patterns in ARs and dynamically select them for each head by computing approximate attention scores during inference. However, these methods still rely on patterns from ARs, and sparse attention in DLMs remains largely unexplored.

**Efficient DLMs** Prior works on accelerating DLMs' inference primarily focuses on cache-based approaches (Ma et al., 2025; Wu et al., 2025). For example, dKV-Cache (Ma et al., 2025) exploits the stability of activations in decoded tokens by caching their key–value states to reduce redundant computation. Fast-dLLM (Wu et al., 2025) further introduces a block-wise caching scheme that caches both prefix and suffix tokens for improved efficiency. While these methods achieve substantial latency reduction, they suffer from noticeable accuracy degradation, especially in long-context scenarios. In this paper, instead of relying on cache-based techniques, we propose a new sparse attention method that reduces inference latency with lossless accuracy.

## 3 METHOD

### 3.1 PRELIMINARY

Diffusion language models (DLMs) generate text via an iterative unmasking process over $T$ discrete denoising steps, gradually transforming a masked sequence into the final output. Formally, let $\mathbb{V}$ denote the vocabulary, and let $\boldsymbol{x}_{:l}^t \in \mathbb{V}^l$ denote the sequence state of length $l$ at step $t$, where $t = 0, \ldots, T$. The initial state is defined as $\boldsymbol{x}_{:l}^T = (c_1, \ldots, c_p, [MASK], \ldots, [MASK])$, where $(c_1, \ldots, c_p)$ represents the prompt (prefill tokens), and the remaining $l - p$ positions are occupied by mask tokens to be generated (generation tokens). Through iterative denoising of both prefill and all generation tokens, DLMs achieve strong performance on language understanding and generation.

However, denoising all tokens across all diffusion steps incurs substantial computational overhead due to the quadratic complexity of the attention mechanism with respect to sequence length $l$. This challenge becomes even more severe in the long-context setting. To tackle this challenge, sparse attention becomes a promising solution. The sparse attention mechanism reduces redundant computation by focusing only on the most important query-key pairs, i.e., important attention scores. The attention score $\boldsymbol{A} \in \mathbb{R}^{l \times l}$ is computed as the scaled dot product between the query matrix $\boldsymbol{Q} \in \mathbb{R}^{l \times d}$ and the key matrix $\boldsymbol{K} \in \mathbb{R}^{l \times d}$, normalized by the square root of the head dimension $d$. Formally, the attention score is defined as:

$$\boldsymbol{A} = A(\boldsymbol{Q}, \boldsymbol{K}) = Softmax(\frac{1}{\sqrt{d}}(\boldsymbol{Q} \cdot \boldsymbol{K}^T)). \tag{1}$$

Each $\boldsymbol{A}_{i,j}$ can be viewed as the dot product between a query–key pair $\boldsymbol{Q}_{i,:}$ and $\boldsymbol{K}_{j,:}$. To improve computational efficiency, sparse attention restricts computation to a subset of query–key pairs. A simple strategy is to retain only the top-$\rho\%$ pairs with the highest scores $\boldsymbol{A}_{i,j}$ for each query $i$ as sparse patterns. The overall index set for selected query-key pairs is defined as

$$\mathbb{S} = \bigcup_{i \in \mathbb{Z}} \mathbb{S}_i = \bigcup_{i \in \mathbb{Z}} \text{Top}_{\rho\%}\{(i,j)|j \in \mathbb{Z}, \text{ranked by } \boldsymbol{A}_{i,j}\}. \tag{2}$$

Then, the sparse attention mechanism is defined as:

$$\boldsymbol{A} = A(\boldsymbol{Q}, \boldsymbol{K}, \boldsymbol{M}_S) = Softmax(\frac{1}{\sqrt{d}}(\boldsymbol{Q} \cdot \boldsymbol{K}^T + \boldsymbol{M}_S)). \tag{3}$$

Here, $\boldsymbol{M}_S$ is a sparse attention pattern based on $\mathbb{S}$, defined as:

$$\boldsymbol{M_S}[i,j] = \begin{cases} 0, & \text{if } (i,j) \in \mathbb{S}, \\ -\infty, & \text{otherwise.} \end{cases} \tag{4}$$

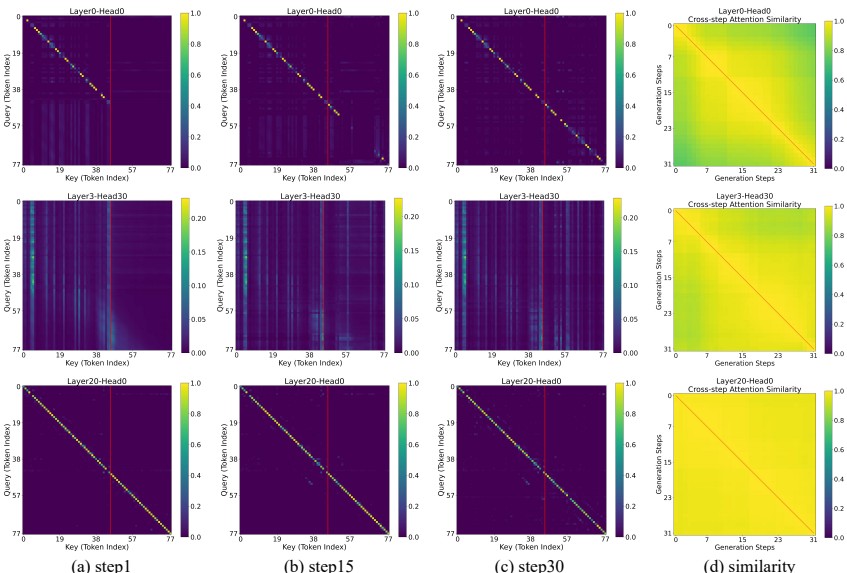

Figure 1: Attention score across denoising steps using LLaDA-1.5 ($l = 78, T = 32, block\_length = 32$). Rows correspond to different attention heads. Red lines divide key tokens in prefill and generation tokens. The result shows pronounced similarity across denoising steps. More visualized attention patterns from different DLMs are provided in the Appendix A.1.

The goal of sparse attention is to minimize the discrepancy between $A(\boldsymbol{Q}, \boldsymbol{K}, \boldsymbol{M}_S) \cdot \boldsymbol{V}$ and $A(\boldsymbol{Q}, \boldsymbol{K}) \cdot \boldsymbol{V}$, where $\boldsymbol{V} \in \mathbb{R}^{l \times d}$ is the value matrix in the attention mechanism. Existing approaches in ARs achieve this goal either by using fixed sparse patterns (Xiao et al., 2023) or by dynamically selecting suitable patterns for each attention head (Lai et al., 2025; Zhang et al., 2025; Xiao et al., 2025). The commonality among these methods is that they all rely on attention patterns observed in ARs. Although the attention mechanism in DLMs also exhibits clear sparse patterns (Figure 1), strategies developed for ARs are not well suited to DLMs. This incompatibility primarily arises from the distinct attention patterns observed in DLMs, as discussed in Section 3.2. To address this issue, we build on these unique observations and propose our method in Section 3.3.

## 3.2 OBSERVATIONS

To enable sparse attention to accelerate DLMs' inference while preserving accuracy, we conduct a systematic analysis of attention patterns, which reveals three fundamental properties:

**Head-Specific Attention Patterns**   In the attention mechanism of DLMs, attention scores vary across heads, as shown in Figure 1(a–c). For example, the second row exhibits a column-wise pattern, while the third row shows a sliding-window pattern. In the first row, the upper part follows a sliding-window structure, whereas the lower part displays a column-wise pattern. Such inconsistencies render widely used sparse attention methods in ARs, such as sliding-window (Jiang et al., 2023) and sink attention (Xiao et al., 2023), unsuitable for DLMs.

**Attention Similarity Across Time**   Although attention scores differ across heads, each of them remain highly consistent across denoising steps. As shown in Figure 1(d), the attention scores in each head exhibit high similarity across steps. Since sparse attention patterns are directly derived from attention scores, this consistency suggests that the sparse attention patterns for each head are also largely stable across steps, motivating the sparse reusing strategy in SparseD, detailed in Section 3.3. More analysis of attention similarity is provided in Appendix A.2.

**Significant Impact of Early Steps on Generation**   Unlike traditional AR models that generate each token sequentially from scratch, DLMs decode all tokens simultaneously. In this process, all tokens undergo denoising across every denoising step. This raises a natural question: do all denois-

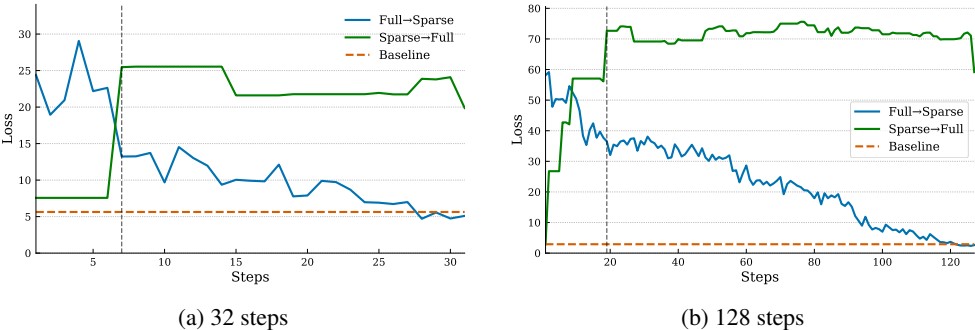

(a) 32 steps                                             (b) 128 steps

Figure 2: Influence of sparse attention across denoising steps. Experiments are conducted on LLaDA-1.5 ($l = 256$ and $block\_length = 256$) with denoising steps 32 and 128. 'Full→Sparse' denotes applying full attention in the first $x$ steps and sparse attention in the remaining steps, while 'Sparse→Full' is the opposite. Sparse attention retains only the top 30% of attention scores per query token (Equation 2). Results highlight the importance of early denoising steps.

ing steps contribute equally? In other words, from a sparse attention perspective, which diffusion steps can apply sparse attention with minimal impact on generation quality?

To investigate this, we evaluate loss changes under different sparse attention configurations (Figure 2). As shown by the green and gray dashed lines, applying sparse attention in the early steps results in a significant loss increase (left side of the gray dashed line), while extending it to additional steps causes only marginal further degradation (right side of the gray dashed line). This indicates that early steps are particularly sensitive to sparse attention. Conversely, the blue line shows that gradually transitioning from sparse to full attention in the early steps substantially reduces loss, further underscoring the critical role of early denoising steps in DLM text generation. These findings demonstrate that directly applying sparse attention methods from ARs in early steps leads to severe degradation in generation quality.

In summary, the above findings show that widely used sparse patterns in ARs fail to capture the head-specific patterns of DLMs, and applying sparse attention to DLMs in the early steps leads to degradation in generation quality. To address these challenges, we propose SparseD for DLMs, based on these unique observations, as discussed in Section 3.3.

### 3.3 SPARSED

At a high level, SparseD is able to efficiently handle the unique attention patterns of DLMs without degrading generation quality. Specifically, SparseD uses full attention in early steps, and then pre-computes and reuses head-specific sparse patterns for sparse attention in subsequent steps. An overview of SparseD is shown in Figure 3, and this section details each of its components.

**Isolated Selection** As discussed in Section 3.2, DLMs exhibit head-specific attention patterns. This makes fixed sparse patterns, e.g., the sliding-window scheme, insufficient to capture important attention scores in DLMs. To address this problem, we compute and select important attention scores for each attention head to form head-specific sparse patterns using Equation 2 and 4. Moreover, since some heads gradually increase attention score on generation tokens in the key dimension (right side of the red line in Figure 1(a–c)), selecting dominant attention scores in the early stages may overlook the contribution from generation tokens. To address this issue, we separately select attention scores for prefill and generation tokens in the key dimension, applying the same selection ratio $\rho\%$ to both. Then, the index set for selected indices can be formulated as:

$$\mathbb{S} = \bigcup_{i \in \mathbb{Z}} \mathbb{S}_i = \bigcup_{i \in \mathbb{Z}} (\mathbb{S}_i^{pre} \bigcup \mathbb{S}_i^{gen}), \tag{5}$$

where $\mathbb{S}_i^{pre}$ and $\mathbb{S}_i^{gen}$ refer to the index set in prefill and generation tokens, respectively. Considering hardware-friendly acceleration, we select important attention scores in a block-wise manner. Specifically, we first apply the average pooling to $\boldsymbol{A}$, formally $\boldsymbol{A}' = \text{avgpool}(\boldsymbol{A}, \text{block\_size}) \in$

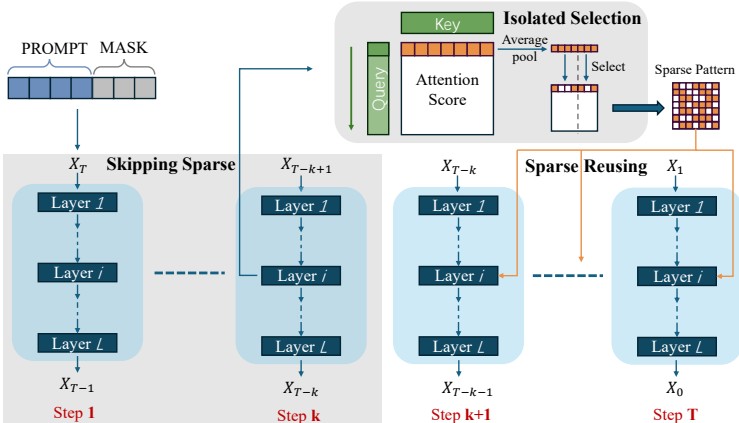

Figure 3: Overview of SparseD. SparseD first applies full attention during the early diffusion steps. It then pre-computes attention scores and selects the important scores using a block-wise scheme, while performing isolated selection for prefill and generation tokens. The resulting sparse patterns are reused in the subsequent steps.

$\mathbb{R}^{l//block\_size \times l//block\_size}$. Then the selecting sets can be formulated as

$$
\begin{aligned}
\mathbb{S}_i^{pre} &= \text{Top}_{\rho\%}\left\{(i,j) \mid 1 \leq j \leq p, \text{ ranked by } \boldsymbol{A}'_{i//block\_size, j//block\_size}\right\}, \\
\mathbb{S}_i^{gen} &= \text{Top}_{\rho\%}\left\{(i,j) \mid p < j \leq l, \text{ ranked by } \boldsymbol{A}'_{i//block\_size, j//block\_size}\right\}.
\end{aligned}
\tag{6}
$$

However, computing the full $A(\boldsymbol{Q}, \boldsymbol{K})$ incurs substantial memory overhead. To address this, we partition $\boldsymbol{Q} \in \mathbb{R}^{l \times d}$ into smaller blocks $\boldsymbol{Q}' \in \mathbb{R}^{block\_size \times d}$ and sequentially compute $\boldsymbol{A}' = \text{avgpool}(A(\boldsymbol{Q}', \boldsymbol{K}), blokc\_size)$ for each block, thereby reducing memory usage.

**Sparse Reusing** As shown in Section 3.2, attention scores within each head show strong similarity over denoising steps. Leveraging this, we can only compute the sparse patterns once and reuse them in subsequent steps. Specifically, we calculate attention scores and select the top-$\rho\%$ important attention score (Equation 5). The resulting selection defines the sparse pattern $M_s$ in Equation 4, which is then reused across denoising steps as shown in Equation 3.

**Skipping Sparse** As discussed in Section 3.2, the early denoising steps are critical for language generation in DLMs, and applying sparse attention at this stage leads to substantial degradation in quality. To address this issue, we apply full attention during the initial $skip\%$ of denoising steps, thereby preserving generation performance. Specifically, full attention is applied during the first $T \times skip\%$ steps. At step $T \times skip\%$, SparseD computes and selects head-specific sparse patterns, which are then reused for sparse attention throughout the remaining $T \times (1 - skip\%)$ steps. In summary, the overview of SparseD is shown in Figure 1 and the pseudo code is shown in Algorithm 1.

---

**Algorithm 1** SparseD

1: **Input:** $\boldsymbol{Q}, \boldsymbol{K}, \boldsymbol{V} \in \mathbb{R}^{l \times d}$, $current\_step$, $T$, $\rho\%$, $skip\%$, $block\_size$
2: **if** $current\_step <= T * skip\%$ **then**             ▷ Skipping Sparse
3:   Attention_Output $\leftarrow$ Full_Attention($\boldsymbol{Q}, \boldsymbol{K}, \boldsymbol{V}$)
4:   **if** $current\_step = T * skip\%$ **then**           ▷ Isolated Selection
5:    **for** $i$ in range($l/block\_size$) **do**
6:     $start = i * block\_size, end = (i+1) * block\_size$
7:     $\boldsymbol{Q}' = \boldsymbol{Q}_{start:end,:}$
8:     $\boldsymbol{A}' \leftarrow \text{avgpool}(A(\boldsymbol{Q}', \boldsymbol{K}), block\_size)$
9:     $\mathbb{S} \leftarrow \text{IndexSelection}(\boldsymbol{A}', \rho\%)$           ▷ Equation 5 and 6
10:     $\boldsymbol{M_S}[start:end,:] \leftarrow \mathbb{S}$              ▷ Equation 4
11:    **end for**
12:   **end if**
13: **else**
14:   Attention_Output $\leftarrow A(\boldsymbol{Q}, \boldsymbol{K}, \boldsymbol{M_S}) \cdot \boldsymbol{V}$         ▷ Sparse Reusing
15: **end if**
16: **return** Attention_Output

---

## 4 EXPERIMENTS

### 4.1 EXPERIMENTAL SETTING

**Models:** We evaluate all comparing methods on recent DLMs, including both LLaDA-1.5 (Zhu et al., 2025) and Dream-7B-Instruct (Ye et al., 2025) models. **Baselines:** We compare SparseD against the original models, the widely used sparse attention methods from ARs (Slide Window and StreamingLLM (Xiao et al., 2023)), and efficient DLM methods (dKV-Cache (Ma et al., 2025) and Fast-dLLM (Wu et al., 2025)). **Datasets:** Experiments are conducted on a diverse set of benchmarks, including general language understanding (MMLU (Hendrycks et al., 2021)), mathematical reasoning (GSM8K (Cobbe et al., 2021)), code generation (HumanEval (Chen et al., 2021)), and long-context evaluation (RULER (Hsieh et al., 2024)). We use 5-shot for MMLU, 4-shot for GSM8K, and 0-shot for the other datasets.

**Implementation Details** All experiments were conducted on NVIDIA A800 (80 GB) GPUs. The original DLMs were accelerated with FlashAttention (Dao, 2023). For the sliding-window method, we set the window size ($ws$) to 256 for short-context evaluations (MMLU, GSM8K, HumanEval) and to 2048 or 4096 for RULER with 4k and 8k contexts, respectively. For StreamingLLM, we set the same $ws$ with the sliding-window method and initial 10% key tokens as sink tokens. For dKV-Cache, we set the cache refresh interval to 2 for the LLaDA-1.5 and to 4 for the Dream-7B-Instruct. For Fast-dLLM, we set the threshold of 0.9, and block size to 8 for MMLU and 32 for other datasets. For SparseD, we set $block\_size = 32$ and $\rho = 50\%$ for short-context tasks, and $block\_size = 128$ with $\rho = 30\%$ for RULER. In all SparseD settings, we use $skip = 20\%$. During the early steps employing full attention, we use FlashAttention as the accelerator, and afterward switch to FlexAttention (Dong et al., 2024), which supports customized sparse patterns. We evaluate accuracy across all datasets and measure latency by processing individual input samples of varying lengths from RULER. Details for datasets, models, and methods are provided in the Appendix A.3.

### 4.2 MAIN RESULTS

Table 1: Comprehensive benchmark results on LLaDA-1.5 and Dream-7B-Instruct.

|  | MMLU | GSM8k | HE | RULER-4k | RULER-8k | Avg. |
|---|---|---|---|---|---|---|
| **Dream-7B-Instruct** | 66.42 | 80.74 | 53.05 | 90.13 | 71.79 | 72.42 |
| + dKV-Cache | 66.32 | **80.67** | **54.88** | 81.41 | 55.08 | 67.67 |
| + Fast-dLLM | 65.51 | 78.17 | 48.78 | 81.68 | 55.64 | 65.95 |
| + Slide Window | 63.45 | 70.20 | 34.76 | 41.46 | 34.36 | 48.84 |
| + StreamingLLM | 64.19 | 72.86 | 33.54 | 43.94 | 36.36 | 50.17 |
| + SparseD | **66.34** | 80.29 | 53.05 | **89.76** | **72.47** | **72.38** |
| **LLaDA-1.5** | 64.24 | 80.38 | 40.85 | 90.45 | 60.73 | 67.33 |
| + dKV-Cache | 63.45 | 79.98 | **40.85** | 88.18 | 57.11 | 65.91 |
| + Fast-dLLM | 63.17 | **82.64** | 40.24 | 86.64 | 47.76 | 64.09 |
| + Slide Window | 63.72 | 57.77 | 27.44 | 39.20 | 36.32 | 44.89 |
| + StreamingLLM | 63.52 | 52.01 | 37.20 | 40.39 | 36.62 | 45.94 |
| + SparseD | **64.14** | 79.80 | **40.85** | **90.89** | **62.44** | **67.62** |

This section presents a comparative evaluation of SparseD from accuracy and latency perspectives.

**Accuracy** As shown in Table 1, SparseD achieves lossless performance compared with the original models. On average, it incurs only a $0.04\%$ accuracy drop on Dream-7B-Instruct and even yields a $0.29\%$ improvement on LLaDA-1.5. In contrast, compared with sparse attention methods in ARs, the sliding-window method and StreamingLLM struggle to handle head-specific attention patterns in DLMs, whereas SparseD delivers great performance in maintaining original capacity. The underperformance of StreamingLLM indicates the incompatibility of attention sinks in ARs with DLMs, a conclusion that aligns with the finding in Rulli et al. (2025). Compared with efficient DLM approaches such as dKV-Cache and Fast-dLLM, SparseD shows clear advantages in long-context scenarios. Although cache-based methods perform well on short-context tasks, they experience significant accuracy degradation with long contexts. Compared with SparseD on RULER-8k, both

dKV-Cache and Fast-dLLM show approximately a 16% accuracy reduction on Dream-7B-Instruct. Additionally, they exhibit 5.3% and 14.6% accuracy reductions on LLaDA-1.5, respectively, highlighting their limitations. Detailed accuracy comparisons on the long-context RULER dataset are provided in Table 5 of Appendix A.4. In addition to Dream-7B and LLaDA-1.5, we also evaluate the effectiveness of SparseD on the multimodal DLM, LLaDA-V (You et al., 2025), for a more comprehensive assessment in Appendix A.6.

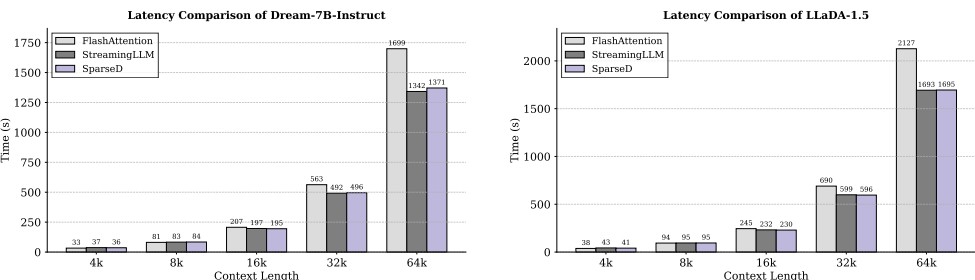

Figure 4: Latency comparison ($T$=128) for Dream-7B-Instruct and LLaDA-1.5, evaluated on a single sample from the RULER dataset with varying sequence lengths.

**Latency**   To evaluate the inference latency of SparseD, we compare it with the sparse attention method (StreamingLLM) and FlashAttention across different context lengths with 128 steps. For StreamingLLM, the window size is set to $ws = \frac{l}{2}$, and sink token length is set to $sink = 10\% \times l$. As shown in Figure 4, SparseD matches FlashAttention at 4k and 8k length, and demonstrates clear advantages beyond 16k length. In particular, at 64k, SparseD achieves $1.23\times$ and $1.25\times$ speedups over FlashAttention on Dream-7B-Instruct and LLaDA-1.5 model, respectively. Although SparseD achieves similar acceleration compared with StreamingLLM, our method maintain accuracy with lossless loss while StreamingLLM lead to great degradation in generation.

Beyond evaluation at 128 diffusion steps, we further assess SparseD under varying numbers of diffusion steps. As shown in Figure 5, the results demonstrate a gradual increase in acceleration compared to FlashAttention. At 128 steps, SparseD achieves $1.23\times$ and $1.25\times$ speedups over FlashAttention on Dream-7B-Instruct and LLaDA-1.5, respectively. At 1024 steps, the speedups increase to $1.50\times$ and $1.48\times$, respectively. This efficiency gain arises because sparse attention patterns are pre-computed only once and reused across all denoising steps. Consequently, in scenarios with long contexts and many diffusion steps, SparseD effectively amortizes the computational cost.

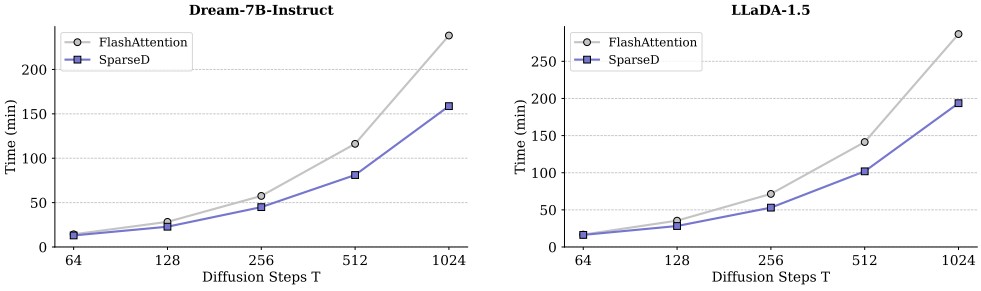

Figure 5: Latency comparison of SparseD on Dream-7B-Instruct and LLaDA-1.5 across varying diffusion steps, evaluated on a single RULER sample with a 64k context length.

### 4.3   ABLATIONS

In this section, we conduct extensive ablation studies to evaluate the components of SparseD and their effects under different configurations. We first validate the effectiveness of its core components—skipping sparse, sparse reusing, and isolated selection. Next, we quantitatively evaluate the similarity of sparse patterns in SparseD. Finally, we report the resource overhead during the precompute stage to demonstrate the efficiency of SparseD.

**Effectiveness of Each Component** As shown in Table 2, we conduct an ablation study to evaluate the effectiveness of each component in SparseD. Removing skipping sparse attention (third row) causes a severe accuracy drop, highlighting its importance in preventing degradation during early steps. Recomputing sparse patterns at every denoising step (fourth row) introduces substantial latency due to repeated computations, whereas reusing sparse patterns (second row) achieves comparable accuracy with far lower latency. Furthermore, excluding isolated selection (last row) decreases accuracy, while enabling it (second row) improves accuracy with negligible latency overhead. In summary, these results collectively confirm that all three components are crucial for making SparseD both effective and efficient for DLMs.

Table 2: Ablation study of SparseD on LLaDA-1.5. Each component is excluded individually to assess its contribution. Accuracy is measured on RULER at 4k length, and latency is evaluated on a 64k-length RULER sample. The relative changes compared to SparseD are highlighted in gray.

| LLaDA-1.5 | RULER (%) | Latency (s) |
|---|---|---|
| FlashAttention | 90.45 | 2127 |
| **SparseD** | **90.89** | **1695** |
| - Skipping Sparse | 87.91 (-3.07%) | 1552 (-8.43%) |
| - Sparse Reusing | 90.82 (-0.07%) | 30020 (+1671%) |
| - Isolated Selection | 90.53 (-0.36%) | 1687 (-0.47%) |

**Quantitative Analysis of Attention Similarity** In Section 3.2, we observe that attention patterns in DLMs exhibit strong temporal similarity. SparseD leverages this property by reusing the sparse attention patterns in the subsequent steps of $T \times skip\%$. To further validate the effectiveness of this strategy, we evaluate the similarity of the computed sparse attention patterns $M_S$ during the reuse phase, i.e., after step $T \times skip\%$.

As shown in Figure 6, we compute the Jaccard similarity of the selected top-$\rho$ blocks between the pattern obtained at step $T \times skip\%$ and the "ground-truth" top-$\rho$ blocks at each subsequent step. The experimental setting is: $T = 32$, $skip = 0.2$, $block\_size = 32$, and $\rho = 0.3$. The results show that the selected blocks across all attention heads achieve an average similarity above 90%, demonstrating the high consistency of the selected blocks throughout the reuse steps.

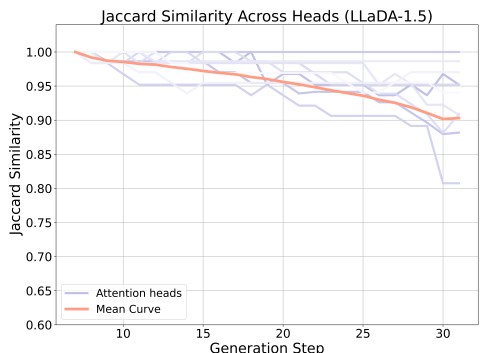

Figure 6: Jaccard similarity across attention heads on LLaDA-1.5. The light blue curves correspond to ten randomly selected attention heads, while the orange curve represents the average similarity across all attention heads.

**Pre-compute Overhead** Table 3 reports the pre-compute overhead, including the storage and latency costs of computing the sparse attention pattern $M_S$, as well as the memory savings enabled by block-wise selection.

Table 3: Pre-compute overhead. 'OOM' represents out of memory.

| LLaDA-1.5 ($T$=128) | 4k | 8k | 16k | 32k | 64k |
|---|---|---|---|---|---|
| Original Memory (MB) | 18709 | 21977 | 29197 | 44989 | 75571 |
| $M_S$ Storage (MB) | 0.48 | 2.84 | 13.37 | 59.10 | 246.11 |
| w/o Block-wise selection (MB) | 21445 | 39065 | OOM | OOM | OOM |
| w/ Block-wise selection (MB) | 18713 | 22031 | 29219 | 44997 | 75583 |
| Original Latency (s) | 38.91 | 94.65 | 245.33 | 690.79 | 2127.72 |
| SparseD Latency (s) | 41.32 | 95.81 | 230.11 | 596.70 | 1695.49 |
| Precompute Time (s) | 3.18 | 6.48 | 17.16 | 60.10 | 233.30 |

The storage cost of $M_S$ arises from storing each pattern as a boolean mask of size $\{0, 1\}^{l/block\_size \times l/block\_size}$. For $Num_{head}$ attention heads across all layers, the total storage amounts to $Num_{head} \cdot l^2/block\_size^2$ boolean entries. As shown in Table 3, the overall storage

of $M_S$ remains modest. Even at 246 MB for a 64k-length setting, it is still negligible compared to the runtime memory (75,583 MB).

As shown in Table 3, pre-computation accounts for roughly 6.3%–13.7% of the total time in SparseD. Since this stage is executed only once, its relative cost diminishes as the number of denoising steps $T$ increases, effectively amortizing the pre-computation overhead.

Block-wise selection effectively reduces the memory cost during the pre-computation of $M_S$. Computing attention scores requires full attention, incurring an $O(l^2)$ memory cost per attention map, which prevents leveraging the memory-efficient implementation of FlashAttention. To mitigate this, we adopt block-wise selection, loading only $O(l \times block\_size)$ attention scores into memory at a time, thereby significantly lowering the memory footprint. As shown in Table 3, without block-wise selection, it can run out of memory for sequences longer than 16k. In contrast, with block-wise selection, SparseD maintains a memory cost comparable to the original FlashAttention implementation.

### 4.4 HYPE-PARAMETER ANALYSIS

In this section, we analyze the two key hyperparameters of SparseD: the skipping ratio ($skip$) and the selection ratio ($\rho$).

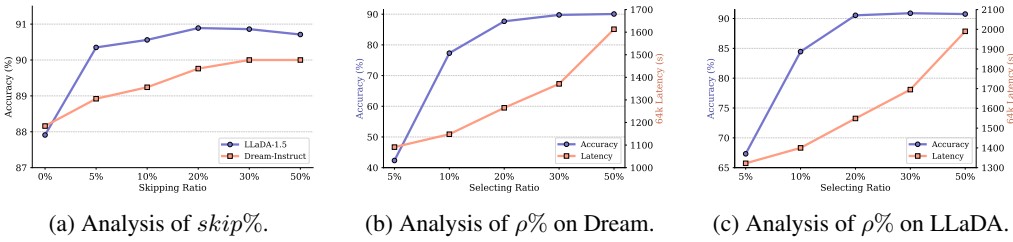

(a) Analysis of $skip\%$.  (b) Analysis of $\rho\%$ on Dream.  (c) Analysis of $\rho\%$ on LLaDA.

Figure 7: Hyper-parameter analysis in Dream-7B-Instruct and LLaDA-1.5 with RULER-4k dataset.

**Skipping Ratio**  As shown in Figure 7a, for both LLaDA-1.5 and Dream-Instruct, increasing the skipping ratio in SparseD improves accuracy. LLaDA-1.5 shows a stronger gain, reaching above 90.89% accuracy at $skip = 20\%$. Dream-Instruct steadily increases and plateaus at 90.00% accuracy once the skipping ratio exceeds $skip = 30\%$. This suggests a moderate skipping ratio (20–30%) achieves the best balance. This result further verifies the observation in Figure 2. For an optimal balance between accuracy and efficiency, we set $skip = 20\%$ in all experiments for both models.

**Selecting Ratio**  For the Dream-7B-Instruct model, as shown in Figure 7b, accuracy rises sharply from around 40% at $\rho = 5\%$ to nearly 89.76% at $\rho = 30\%$, after which it saturates. In contrast, latency increases steadily with higher $\rho$. A similar trend is observed for LLaDA-1.5 in Figure 7c, with accuracy increasing until $\rho = 20\%$ and then saturating. To achieve a balanced trade-off between accuracy and efficiency, we set $\rho = 30\%$ for long-context experiments.

## 5 CONCLUSIONS

In this paper, we propose a novel sparse attention method, **SparseD**, for DLMs. The design of SparseD is based on three key observations in DLMs: (1) attention patterns vary across attention heads, showing head-specific patterns, (2) attention patterns remain highly consistent across denoising steps, and (3) early diffusion steps are crucial for effective language generation. Leveraging these insights, SparseD pre-computes sparse attention patterns for each head once and reuses them across diffusion steps. Additionally, SparseD applies full attention in the early steps and skips sparse attention to preserve generation quality. These designs enable SparseD to efficiently handle head-specific patterns while avoiding degradation in generation quality, making it a practical and effective solution for deploying DLMs in long-context applications. Extensive experiments demonstrate that SparseD achieves lossless performance on all tested benchmarks while delivering up to 1.50× speedup over FlashAttention at a 64k context length with 1,024 diffusion steps.

## 6 Acknowledgments

This project is supported by the Ministry of Education, Singapore, under its Academic Research Fund Tier 2 (Award Number: MOE-T2EP20122-0006) and the Hong Kong Polytechnic University under the Presidential Young Scholars Scheme (Project ID: P0058232).

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

# A APPENDIX

## A.1 ATTENTION PATTERNS

In this section, we visualize a broader range of attention patterns in DLMs to further support the observations discussed in Section 3.2. Specifically, the attention patterns of LLaDA-8B-Base and Dream-7B-Instruct are shown in Figures 8 and 9, respectively. As illustrated in Figures 8(a–c) and 9(a–c), both models display distinct head-specific attention patterns. Moreover, Figures 8(d) and 9(d) show strong consistency across denoising steps. These findings align well with the observations in Section 3.2.

Admittedly, certain corner cases exist. For example, the last rows of Figures 8(d) and 9(d) reveal instances with reduced similarity across steps. In Figure 8(d), although the early steps deviate, the later steps still maintain strong similarity. In Figure 9(d), the similarity appears in a block-wise manner; nevertheless, it still remains above 60% similarity across steps. Moreover, such cases are rare among all attention heads.

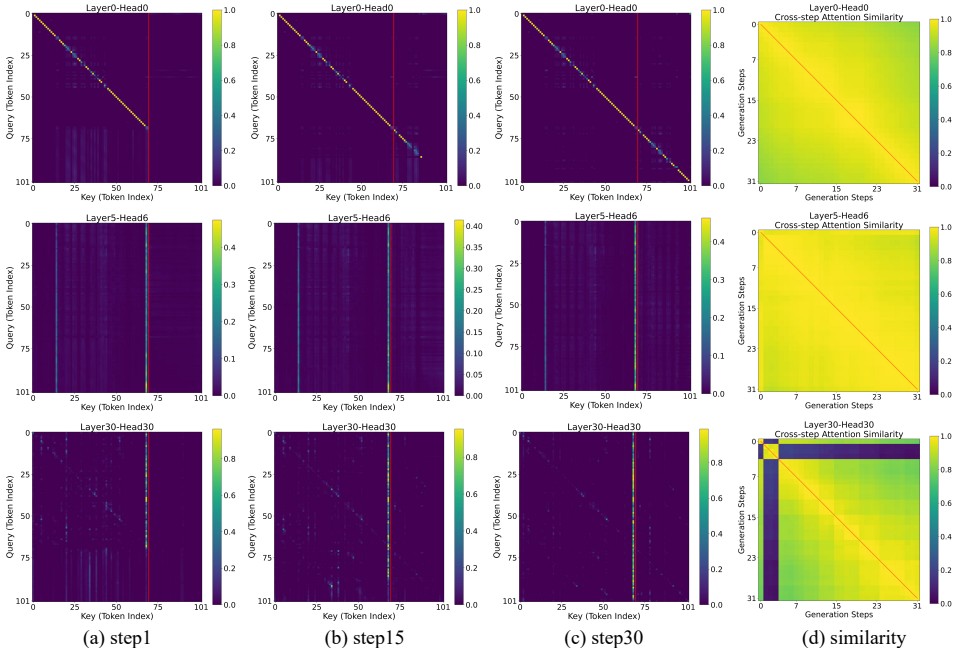

(a) step1      (b) step15      (c) step30      (d) similarity

Figure 8: Attention score across denoising steps using LLaDA-8B-Base ($l = 102$, $T = 32$, $block\_length = 32$). Rows correspond to different attention heads. Red lines divide key tokens in prefill and generation tokens. The result shows pronounced similarity across denoising steps.

## A.2 ANALYSIS OF ATTENTION SIMILARITY

This section provides a more detailed analysis of the strong similarity observed in the attention maps of DLMs. In Section 3.2, we demonstrated this phenomenon by directly measuring the cosine similarity of the attention scores $A$ across denoising steps for each attention head. Since $A$ is computed as $QK^\top$ (Equation 1), the high similarity of $A$ may originate from the intrinsic temporal consistency of the query ($Q$) and key ($K$) representations. Therefore, in this section, we further analyze the cross-step similarity of both $Q$ and $K$.

As shown in Figure 10, we visualize the cross-step cosine similarity matrices of the attention scores, queries, and keys across different layers and heads of LLaDA-1.5. Across all three components—attention scores (column a), queries (column b), and keys (column c)—we observe consistently high temporal similarity. The high similarity between $Q$ and $K$ provides evidence that the similarity in attention scores primarily originates from $Q$ and $K$.

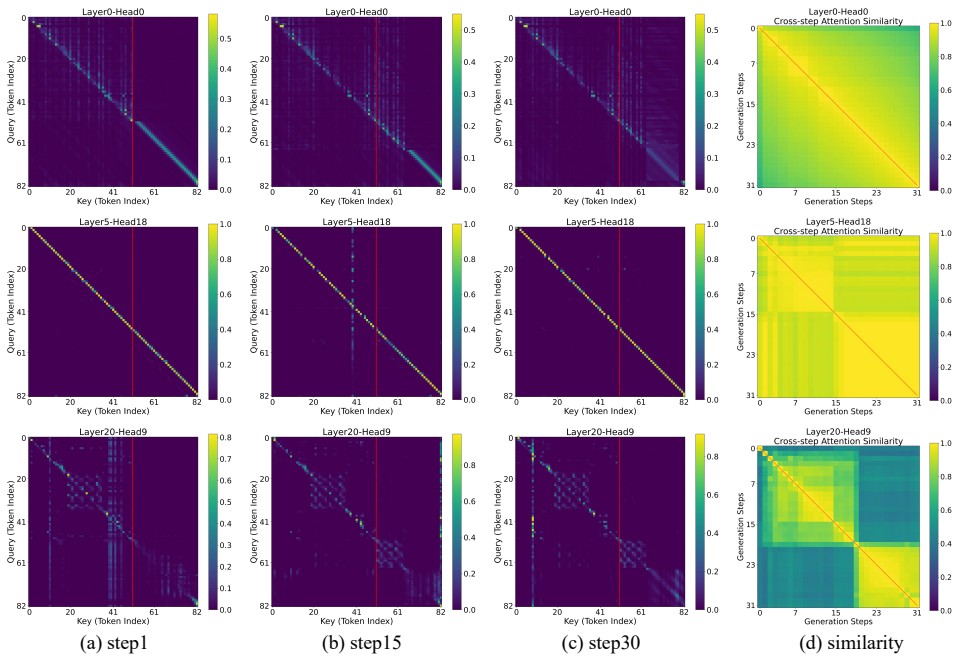

Figure 9: Attention score across denoising steps using Dream-7B-Instruct ($l = 83$, $T = 32$). Rows correspond to different attention heads. Red lines divide key tokens in prefill and generation tokens. The result shows pronounced similarity across denoising steps.

Notably, in rows 1 and 2, $\boldsymbol{Q}$ and $\boldsymbol{K}$ exhibit similarity trends that closely mirror those of $\boldsymbol{A}$, indicating that the stability of attention scores stems from the consistency of the underlying query and key representations. In row 3, although the query and key do not follow the same similarity pattern, their product—the attention scores—still exhibits high similarity.

## A.3  EXPERIMENTAL DETAILS

In this section, we provide the detailed evaluation settings described in Section 4. Specifically, we report the parameters used for different models, methods, and datasets.

**Datasets**  Experiments are conducted on a diverse set of benchmarks, including general language understanding (MMLU (Hendrycks et al., 2021)), mathematical reasoning (GSM8K (Cobbe et al., 2021)), code generation (HumanEval (Chen et al., 2021)), and long-context evaluation (RULER (Hsieh et al., 2024)).

The RULER dataset consists of 13 subtasks for comprehensive evaluation, including *niah_single_1*, *niah_single_2*, *niah_single_3*, *niah_multikey_1*, *niah_multikey_2*, *niah_multikey_3*, *niah_multiquery*, *niah_multivalue*, *ruler_vt*, *ruler_fwe*, *ruler_qa_squad*, and *ruler_qa_hotpot*. Depending on the specific subtask, we set different evaluation parameters—particularly $T$ and $l$—to enable efficient evaluation. The details are shown in the Table 4.

**Methods**  We compare SparseD with the slide-window approach, StreamingLLM, dKV-Cache, and Fast-dLLM. For SparseD, the slide-window method and StreamingLLM. Details of SparseD, the slide-window approach, and StreamingLLM are shown in Table 4. Note that in StreamingLLM, $sink$ refers to the ratio of initial tokens designated as sink tokens. For the slide-window approach and StreamingLLM, we use FlexAttention for acceleration.

For dKV-Cache, we employ the dKV-Cache-PD variant on the Dream-7B-Instruct model, setting the cache refresh interval to 4. For the LLaDA-1.5 model, we adopt the dKV-Cache-Greedy variant, with a cache refresh interval of 2 and a window size of 4. For Fast-dLLM, we use the Prefix KV

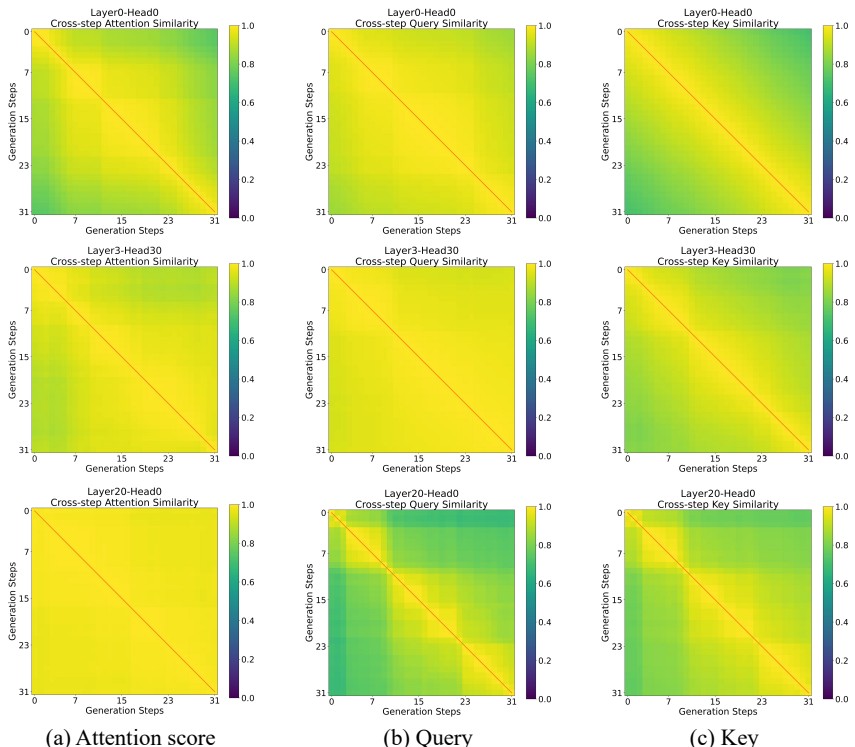

(a) Attention score        (b) Query        (c) Key

Figure 10: The similarity analysis of attention score, query, and key. The experiment is conducted on LLaDA-1.5 (setting the same as Figure 1).

Cache version, setting the threshold to 0.9, with a block size of 8 for MMLU and 32 for the other datasets.

## A.4    EVALUATION DETAILS ON RULER DATASET

Since we use different configurations for evaluating the subtasks of the RULER dataset, in this section, we present the detailed accuracy results for each subtask. Specifically, Table 5 provides the detailed breakdown corresponding to Table 1, while Table 6 presents the detailed results corresponding to Table 2.

## A.5    DISCUSSION WITH CACHE-BASED METHODS

In this section, we provide a discussion of the relationship between SparseD and cache-based acceleration methods. Although SparseD and cache-based approaches belong to two fundamentally different categories, both aim to improve the efficiency of DLMs. In Section 4.2, we primarily compare their accuracy to demonstrate the lossless performance of SparseD.

However, a direct comparison of latency between the two is inherently unfair and conceptually mismatched. Cache-based methods accelerate inference by storing and reusing the key-value (KV) representations generated in previous steps. In contrast, SparseD concentrates on sparsifying the attention computation, without relying on any form of caching. Consequently, the two approaches operate under fundamentally different computational assumptions and incur distinct workloads.

For a fair comparison, we combine SparseD with Prefill-Cache (Ma et al., 2025), i.e., caching prompts, to compare with other cache-based methods, dKV-Cache and Fast-dLLM. As shown in Table 7, integrating SparseD with caching can further accelerate inference but causes a performance drop, indicating a degree of incompatibility between the two approaches. In fact, since sparse attention and KV-cache methods are fundamentally different and our work mainly focuses on sparse attention, such incompatibility is expected—especially in the context of DLMs, which remain un-

Table 4: Parameters of Evaluation. 'BS' represents the batch size. For the RULER dataset with varying lengths, certain parameters ($ws$, BS) are configured differently, with specific changes indicated in parentheses to denote the version being used.

| | MMLU | GSM8k | HE | niah_single_1
niah_single_2
niah_multikey_1
niah_multikey_2
ruler_vt
ruler_qa_squad
ruler_qa_hotpot | niah_single_3
niah_multikey_3
niah_multiquery
niah_multivalue
ruler_fwe | ruler_cwe |
|---|---|---|---|---|---|---|
| **Dream-7B
-Instruct** | few_shot=5
BS=1
$T$=8
$l$=8
temperature=0.1
top_p=0.9 | few_shot=4
BS=1
$T$=256
$l$=256
temperature=0.1
top_p=0.9 | few_shot=0
BS=1
$T$=512
$l$=512
temperature=0.1
top_p=0.9 | few_shot=0
BS=16(4k)/8(8k)
$T$=64
$l$=32
temperature=0.1
top_p=0.9 | few_shot=0
BS=16(4k)/8(8k)
$T$=64
$l$=64
temperature=0.1
top_p=0.9 | few_shot=0
BS=16(4k)/8(8k)
$T$=128
$l$=128
temperature=0.1
top_p=0.9 |
| + Slide Window | $ws$=256 | $ws$=256 | $ws$=256 | $ws$=2048(4k)
$ws$=4096(8k) | $ws$=2048(4k)
$ws$=4096(8k) | $ws$=2048(4k)
$ws$=4096(8k) |
| + StreamingLLM | $ws$=256

$sink$=20%*$ws$ | $ws$=256

$sink$=10%*$l$ | $ws$=256

$sink$=10%*$l$ | $ws$=2048(4k)
$ws$=4096(8k)
$sink$=10%*$l$ | $ws$=2048(4k)
$ws$=4096(8k)
$sink$=10%*$l$ | $ws$=2048(4k)
$ws$=4096(8k)
$sink$=10%*$l$ |
| + SparseD | $block\_size$=32
$\rho = 0.5$
$skip = 20\%$ | $block\_size$=32
$\rho = 0.5$
$skip = 20\%$ | $block\_size$=32
$\rho = 0.5$
$skip = 20\%$ | $block\_size$=128
$\rho = 0.3$
$skip = 20\%$ | $block\_size$=128
$\rho = 0.3$
$skip = 20\%$ | $block\_size$=128
$\rho = 0.3$
$skip = 20\%$ |
| **LLaDA-1.5** | few_shot=5
BS=1
$T$=8
$l$=8
block_length=8 | few_shot=4
BS=1
$T$=256
$l$=256
block_length=32 | few_shot=0
BS=1
$T$=512
$l$=512
block_length=32 | few_shot=0
BS=1
$T$=32
$l$=32
block_length=32 | few_shot=0
BS=1
$T$=64
$l$=64
block_length=64 | few_shot=0
BS=1
$T$=128
$l$=128
block_length=128 |
| + Slide Window | $ws$=256 | $ws$=256 | $ws$=256 | $ws$=2048(4k)
$ws$=4096(8k) | $ws$=2048(4k)
$ws$=4096(8k) | $ws$=2048(4k)
$ws$=4096(8k) |
| + StreamingLLM | $ws$=256

$sink$=20%*$ws$ | $ws$=256

$sink$=10%*$l$ | $ws$=256

$sink$=10%*$l$ | $ws$=2048(4k)
$ws$=4096(8k)
$sink$=10%*$l$ | $ws$=2048(4k)
$ws$=4096(8k)
$sink$=10%*$l$ | $ws$=2048(4k)
$ws$=4096(8k)
$sink$=10%*$l$ |
| + SparseD | $block\_size$=32
$\rho = 0.5$
$skip = 20\%$ | $block\_size$=32
$\rho = 0.5$
$skip = 20\%$ | $block\_size$=32
$\rho = 0.5$
$skip = 20\%$ | $block\_size$=128
$\rho = 0.3$
$skip = 20\%$ | $block\_size$=128
$\rho = 0.3$
$skip = 20\%$ | $block\_size$=128
$\rho = 0.3$
$skip = 20\%$ |

Table 5: Evaluation details of RULER in main results (Table 1).

| | niah | | | | | | | | ruler | | | | | |
| | S_1 | S_2 | S_3 | MK_1 | MK_2 | MK_3 | MQ | MV | VT | CWE | FWE | QS | QH | AVG |
|---|---|---|---|---|---|---|---|---|---|---|---|---|---|---|
| **4K-Length** | | | | | | | | | | | | | | |
| **Dream-7B-Instruct** | 100.00 | 100.00 | 93.00 | 98.60 | 99.80 | 79.80 | 97.45 | 98.75 | 96.36 | 87.12 | 78.73 | 78.68 | 63.40 | 90.13 |
| + dKV-Cache | 100.00 | 98.88 | 81.40 | 83.40 | 99.60 | 44.00 | 87.10 | 82.10 | 91.12 | 72.48 | 75.40 | 78.75 | 64.20 | 81.41 |
| + Fast-dLLM | 100.00 | 98.20 | 78.00 | 83.40 | 99.60 | 44.20 | 88.25 | 88.75 | 91.00 | 73.58 | 74.47 | 78.62 | 63.80 | 81.68 |
| + Slide Window | 26.00 | 29.40 | 28.20 | 30.40 | 28.40 | 27.30 | 26.80 | 27.30 | 36.84 | 86.60 | 80.33 | 79.53 | 46.80 | 41.46 |
| + StreamingLLM | 32.00 | 29.60 | 28.80 | 31.00 | 28.80 | 22.20 | 27.90 | 28.45 | 36.84 | 86.78 | 88.13 | 81.98 | 48.80 | 43.94 |
| + SparseD | 100.00 | 100.00 | 93.60 | 97.40 | 99.60 | 79.60 | 97.00 | 97.15 | 96.80 | 83.50 | 80.07 | 79.22 | 63.00 | 89.76 |
| **LLaDA-1.5** | 100.00 | 100.00 | 100.00 | 100.00 | 100.00 | 100.00 | 95.05 | 99.80 | 100.00 | 56.24 | 63.80 | 83.17 | 77.80 | 90.45 |
| + dKV-Cache | 100.00 | 100.00 | 96.60 | 100.00 | 100.00 | 96.20 | 99.40 | 94.20 | 96.60 | 50.86 | 56.67 | 80.47 | 75.40 | 88.18 |
| + Fast-dLLM | 100.00 | 100.00 | 88.40 | 100.00 | 100.00 | 95.60 | 99.95 | 98.05 | 88.16 | 45.32 | 51.13 | 83.00 | 76.80 | 86.64 |
| + Slide Window | 25.60 | 29.40 | 28.20 | 30.40 | 28.20 | 42.00 | 24.35 | 24.05 | 38.64 | 55.98 | 52.13 | 80.30 | 50.40 | 39.20 |
| + StreamingLLM | 25.80 | 29.40 | 28.20 | 30.40 | 28.20 | 42.00 | 27.70 | 27.30 | 38.68 | 38.62 | 73.87 | 82.30 | 52.60 | 40.39 |
| + SparseD | 100.00 | 100.00 | 100.00 | 100.00 | 100.00 | 100.00 | 100.00 | 98.60 | 100.00 | 53.74 | 68.87 | 83.40 | 77.00 | 90.89 |
| **8K-Length** | | | | | | | | | | | | | | |
| **Dream-7B-Instruct** | 99.80 | 90.80 | 67.20 | 75.40 | 71.80 | 28.40 | 93.10 | 92.70 | 63.08 | 57.50 | 79.27 | 58.45 | 55.80 | 71.79 |
| + dKV-Cache | 89.20 | 71.20 | 40.00 | 48.40 | 71.00 | 16.20 | 62.45 | 47.90 | 32.00 | 49.08 | 76.53 | 56.55 | 55.60 | 55.08 |
| + Fast-dLLM | 89.00 | 69.80 | 39.20 | 47.60 | 71.40 | 15.60 | 67.25 | 58.65 | 30.68 | 45.64 | 76.27 | 56.68 | 55.60 | 55.64 |
| + Slide Window | 26.80 | 29.60 | 28.60 | 31.40 | 20.40 | 21.90 | 29.45 | 28.05 | 30.84 | 61.34 | 61.93 | 27.97 | 48.40 | 34.36 |
| + StreamingLLM | 37.00 | 29.60 | 28.60 | 33.00 | 20.40 | 21.20 | 29.95 | 28.70 | 31.48 | 61.04 | 72.53 | 29.70 | 49.60 | 36.36 |
| + SparseD | 99.80 | 91.40 | 71.60 | 76.00 | 72.60 | 28.40 | 93.55 | 93.75 | 65.00 | 57.42 | 79.53 | 56.88 | 56.20 | 72.47 |
| **LLaDA-1.5** | 63.00 | 71.40 | 58.20 | 64.40 | 55.80 | 41.40 | 69.00 | 64.55 | 63.96 | 67.12 | 61.60 | 49.95 | 59.20 | 60.73 |
| + dKV-Cache | 63.20 | 74.40 | 54.60 | 64.60 | 56.60 | 27.60 | 59.15 | 59.60 | 54.84 | 64.92 | 55.13 | 48.87 | 59.00 | 57.11 |
| + Fast-dLLM | 53.80 | 66.60 | 40.80 | 58.20 | 53.80 | 35.40 | 54.50 | 45.25 | 29.96 | 26.50 | 45.47 | 50.92 | 59.80 | 47.76 |
| + Slide Window | 26.20 | 29.60 | 28.60 | 31.80 | 26.00 | 20.20 | 30.40 | 28.15 | 38.60 | 57.94 | 71.47 | 27.27 | 56.00 | 36.32 |
| + StreamingLLM | 26.20 | 29.60 | 28.60 | 31.80 | 26.00 | 20.20 | 30.45 | 28.15 | 38.64 | 56.62 | 76.93 | 27.37 | 55.60 | 36.62 |
| + SparseD | 74.00 | 75.20 | 58.40 | 67.20 | 60.60 | 41.60 | 69.50 | 62.35 | 66.60 | 57.44 | 68.60 | 50.68 | 59.60 | 62.44 |

derexplored. In future work, we will investigate how to better integrate these methods to achieve lossless acceleration.

Table 6: Evaluation details of RULER-4k in ablation study (Table 2).

| LLaDA-1.5 | S_1 | S_2 | S_3 | niah MK_1 | MK_2 | MK_3 | MQ | MV | VT | CWE | ruler FWE | QS | QH | AVG |
|---|---|---|---|---|---|---|---|---|---|---|---|---|---|---|
| **FlashAttention** | 100.00 | 100.00 | 100.00 | 100.00 | 100.00 | 100.00 | 95.05 | 99.80 | 100.00 | 56.24 | 63.80 | 83.17 | 77.80 | 90.45 |
| SparseD | 100.00 | 100.00 | 100.00 | 100.00 | 100.00 | 100.00 | 100.00 | 98.60 | 100.00 | 53.74 | 68.87 | 83.40 | 77.00 | 90.89 |
| - Skipping Sparse | 100.00 | 100.00 | 100.00 | 100.00 | 100.00 | 100.00 | 99.95 | 95.95 | 97.20 | 34.02 | 59.80 | 81.43 | 74.60 | 87.91 |
| - Sparse Reusing | 100.00 | 100.00 | 100.00 | 100.00 | 100.00 | 100.00 | 100.00 | 98.50 | 100.00 | 53.26 | 68.60 | 83.60 | 76.80 | 90.82 |
| - Isolated Selection | 100.00 | 100.00 | 100.00 | 100.00 | 100.00 | 99.00 | 99.95 | 97.65 | 98.88 | 53.88 | 68.53 | 82.47 | 76.60 | 90.53 |

Table 7: Comparison with cache-based methods. 'OOM' refers to out of memory.

| | Accuracy (%) RULER-4k | Latency (s) 4k | 8k | 16k | 32k | 64k |
|---|---|---|---|---|---|---|
| **LLaDA-1.5** | 90.45 | 38 | 94 | 245 | 690 | 2127 |
| + dKV-Cache | 88.18 | 28 | 64 | 154 | 409 | OOM |
| + Fast-dLLM | 86.64 | 9 | 14 | 17 | OOM | OOM |
| + SparseD + Prefill Cache | 77.42 | 13 | 19 | 32 | 86 | OOM |

## A.6 EVALUATION ON LLaDA-V

Table 8: Evaluation on LLaDA-V.

| | MMMU val | MMMU-Pro standard | ChartQA |
|---|---|---|---|
| **LLaDA-V** | 48.67 | 35.20 | 78.32 |
| + SparseD | 48.56 | 34.39 | 78.12 |

To extend the evaluation to a broader range of models (excluding Dream-7B and LLaDA-1.5), we tested our method on the multimodal DLM, LLaDA-V (You et al., 2025). As shown in Table 8, we compared the results between the original version and the SparseD version. We evaluated them on the MMMU (Yue et al., 2024), MMMU-Pro (Yue et al., 2025), and ChartQA (Masry et al., 2022) benchmarks. For SparseD, we used $\rho = 0.3$ and $skip = 20\%$. The setup for all benchmarks strictly follows You et al. (2025).

The results demonstrate that our method achieves lossless performance on three benchmarks. This further confirms the effectiveness of our approach, extending its success beyond text-only DLMs.

## A.7 LIMITATIONS

This work presents a novel sparse attention method for DLMs that efficiently adapts to head-specific patterns and avoids generation degradation in early denoising steps. However, several limitations remain. One primary limitation of this work lies in its focus on algorithmic design. Future work could explore further system-level optimizations, both for computing sparse attention patterns and for accelerating head-specific sparse patterns. Another aspect is combining our method, a sparse attention-based approach, with cache-based methods. The key to implementation is how to retain the advantages of both the lossless sparse attention method and the fast cache-based method simultaneously.

## A.8 THE USE OF LARGE LANGUAGE MODELS (LLMS)

In our work, Large Language Models (LLMs) are primarily employed to support the writing process, particularly during paper refinement. Specifically, we use LLMs to improve grammar, correct word usage, and polish overall expression. This ensures that our manuscripts maintain both clarity and fluency, making the presentation of our ideas more precise and professional. In contrast, the research ideas and experimental design of this paper were conceived entirely by the authors without the use of LLMs.

