# OpenReview forum: "SparseD: Sparse Attention for Diffusion Language Models"
_ICLR.cc/2026/Conference — ICLR 2026 Poster_

### Official Review · Reviewer_Q4L3 · 2025-10-24

**Soundness:** 3
**Presentation:** 3
**Contribution:** 3
**Rating:** 6
**Confidence:** 3

**Summary:**

This paper addresses the high inference latency of diffusion language models (DLMs), which stems from the quadratic complexity of the attention mechanism applied over all tokens at each denoising step. The authors propose SparseD, a sparse attention method tailored for DLMs. The method is motivated by three key empirical observations: (1) DLM attention patterns are "head-specific," meaning they vary significantly across different attention heads; (2) patterns *within* a given head are "highly similar" across all denoising steps; and (3) the early denoising steps are "critical" for generation quality, and applying sparsity too early causes performance degradation.

Based on these findings, SparseD's design is threefold:
1.  **Skipping Sparse:** It applies full (dense) attention for an initial percentage of denoising steps (e.g., first 20%) to preserve generation quality.
2.  **Isolated Selection:** At the end of this initial phase, it computes the full attention matrix *once* and selects a head-specific, block-wise top-$\rho\%$ sparse pattern. This selection is done "isolating" prefill and generation tokens to ensure both are adequately represented.
3.  **Sparse Reusing:** This single, pre-computed head-specific sparse pattern is then cached and reused for all remaining denoising steps, leveraging the observed temporal stability.

Experiments on models like Dream-7B and LLaDA-1.5 show that SparseD achieves "lossless" accuracy on various benchmarks (MMLU, RULER) while delivering significant speedups over standard FlashAttention (e.g., up to 1.50x at 64k context with 1024 steps), as the one-time pre-computation cost is amortized.

**Strengths:**

1.  **Novel Empirical Insights:** The paper's main strength is its clear identification and empirical validation of three specific attention behaviors in DLMs: (1) head-specific patterns, (2) temporal stability of patterns, and (3) high sensitivity of early denoising steps. This analysis is novel and provides a solid foundation for future work on DLM efficiency.
2.  **Well-Motivated Design:** The SparseD method is a strong example of principled system design. Each of its three components (Skipping Sparse, Isolated Selection, Sparse Reusing) is a direct and logical solution to one of the three identified empirical observations.
3.  **Strong Experimental Validation:** The evaluation is comprehensive. The authors demonstrate that SparseD achieves its "lossless" claim by benchmarking against the original models. Crucially, they also show its superiority to two distinct and relevant classes of baselines: (a) AR-based sparse attention (which fails on accuracy) and (b) other DLM acceleration methods (which degrade on long-context tasks).
4.  **Good Ablation Study:** The ablation in Table 2 is a model of clarity. It individually removes each component of SparseD and shows the precise, and severe, negative impact: removing "Skipping Sparse" destroys accuracy (Observation 3 validated), removing "Sparse Reusing" explodes latency (Observation 2 validated), and removing "Isolated Selection" hurts accuracy (heuristic validated).
5.  **Practicality and Scalability:** The method provides a practical speedup (up to 1.5x) that compellingly scales with the number of denoising steps $T$ (Figure 5). This makes it particularly attractive for high-quality generation, which often requires many steps.

**Weaknesses:**

1.  **Lack of Deeper Analysis:** The paper is purely empirical. It does not offer any hypothesis or theoretical exploration for *why* these attention patterns are temporally stable. The denoising process is defined by $T$ steps of iterative refinement, so the $Q$, $K$, and $V$ representations are constantly changing. It is non-obvious and fascinating why the resulting $QK^T$ patterns should remain static. While a full theoretical proof is not required, some discussion or preliminary analysis (e.g., measuring the cosine similarity of $Q$/$K$ vectors over time) would have strengthened the paper.
2.  **Scalability of the $O(N^2)$ Pre-computation Step:** The method requires a single, full $O(N^2)$ attention computation at step $T \times \text{skip}\%$ to build the sparse pattern. While the authors use a memory-efficient block-wise implementation, the *computational cost* of this step is still $O(N^2)$. The experiments stop at 64k context. For truly massive contexts (e.g., 256k, 1M), this single pre-computation step could become the new latency bottleneck, potentially dominating the cost of all subsequent sparse steps. The paper does not analyze the scaling of this specific step or discuss this potential limitation.

**Questions:**

1.  Could the authors provide a more quantitative analysis of "Attention Similarity Across Time" (Observation 2)? For example, by plotting the Jaccard similarity of the selected top-$\rho\%$ blocks between the pattern computed at step $T \times \text{skip}\%$ and the "ground truth" top-$\rho\%$ blocks at each subsequent step $t$? This would be much stronger evidence than the current heatmaps.
2.  How robust is the "lossless" claim to the temporal similarity assumption? What is the performance impact (e.g., on RULER) if the sparse pattern from one prompt is (incorrectly) reused for a different, unseen prompt? This would stress-test the model's reliance on the exact pattern.
3.  Regarding the $O(N^2)$ pre-computation step: How does the wall-clock latency of this *single step* scale with context length (e.g., from 4k to 64k)? At what context length $N$ (assuming $T=1024$) would the authors project this one-time $O(N^2)$ cost to become larger than the cumulative $O(T \cdot N \cdot \rho N)$ cost of the remaining sparse steps?
4.  The "Isolated Selection" heuristic uses the same selection ratio $\rho\%$ for both prefill and generation tokens. Is this an optimal choice? Have the authors experimented with different ratios (e.g., $\rho_{\text{prefill}}$ and $\rho_{\text{gen}}$)? One might hypothesize that static prefill tokens and dynamic generation tokens could benefit from different sparsity levels.

---

> ### Author Response · Authors · 2025-11-22
> **Author Response**
>
> `>>>Q1` **Lack of Deeper Analysis.**
>
> `>>>A1` Thank you for your suggestion. As suggested, we add a similarity analysis of the Q and K matrices in `Appendix A.2`. Our experiments show that the high similarity observed in the attention scores is largely attributed to the strong temporal consistency of the Q and K matrices. The detailed analysis and the corresponding figure can be found in `Appendix A.2`.
>
>
>
> `>>>Q2` **Scalability of the $O(N^2)$ Pre-computation Step.**
>
> `>>>A2` Thank you for the in-depth question. You are correct that block-wise selection is for memory efficiency and still incurs an $O(N^2)$ computational cost. Although pre-computation latency becomes a new bottleneck in long-context settings, the overall latency reduction achieved by our method still exceeds the additional pre-computation overhead.
>
> As `Table 3` shows, pre-computation accounts for roughly 6.3%–13.7% of the total time in SparseD across different sequence lengths. Since this stage is executed only once, its relative cost diminishes as the number of denoising steps $T$ increases, effectively amortizing the pre-computation overhead.
>
>
>
> `>>>Q3` **More quantitative analysis**
>
> `>>>A3` We sincerely appreciate the suggestion. Following your advice, we add a Jaccard similarity analysis in `Section 4.3 (lines 449–466)`. The results show that the blocks selected at step $T \times \text{skip}$% exhibit high consistency with those selected in subsequent steps, with similarity **exceeding 90%**. This finding confirms the strong temporal stability of attention patterns and further demonstrates the effectiveness of our method.
>
>
>
> `>>>Q4` **Robustness**
>
> `>>>A4` Thank you for the question. Using the same sparse pattern across different prompts indeed leads to a significant accuracy drop. **Importantly**, our method does not rely on any pre-computed or fixed sparse pattern before inference; instead, it computes the sparse pattern dynamically for each prompt during inference, allowing the pattern to adapt to the prompt’s content and ensuring lossless performance.
>
> To validate your scenario and verify our conclusion, we conducted an experiment where we applied the sparse pattern extracted from the longest RULER-4k prompt to all other prompts. The results show that this `Incorrect Pattern` manner causes substantial performance degradation, confirming that dynamic, per-prompt pattern selection is crucial for preserving accuracy.
> |LLaDA-1.5|RULER-4k $\uparrow$|
> |-|-|
> |Incorrect Pattern|62.69|
> |SparseD|**90.86**|
>
>
>
> `>>>Q5` **Regarding the $O(N^2)$ Complexity**
>
> `>>>A5` Thanks for raising this point! You are correct that the pre-computation step has $O(N^2)$ complexity. However, in practice, this cost does **not** offset the acceleration achieved by sparse attention.
>
> - **Pre-computation Cost is small**. We added a *new analysis* in `Table 3 (Section 4.3)` showing how the one-time pre-computation latency scales with context length. Across different sequence lengths, pre-computation accounts for **only 6.3%–13.7%** of SparseD’s total time. While this introduces some overhead, it is far outweighed by the overall latency reduction and can be further amortized over multiple diffusion steps.
> - **Pre-computation cost is not larger than cumulative cost**. The concern that pre-computation might exceed the cumulative cost of the remaining sparse steps depends on the **number of denoising steps $T$**, not on the context length. Specifically, the one-time cost exceeds the cumulative cost only when
> $$N^2 > T \cdot N^2 \cdot \rho, \text{or equivalently } \frac{1}{T}>\rho$$
> where $\rho$ is the sparsity ratio. This condition occurs only when $\rho$ is extremely small or $T$ is very small. Both are rare in practical settings.
>
> We will revise the paper to elaborate on this point.
>
>
> `>>>Q6` **Different sparsity ratio in selection**
>
> `>>>A6` Great question! Prefill tokens and generation tokens may favor different sparsity ratios, and using the same ratio for both is not necessarily optimal. We chose a uniform ratio for both to control variables, making our experiments easier to interpret and compare.
>
> To explore this, we conducted a **new experiment** on `HumanEval` using the LLaDA-1.5 model. The results indicate that a 30%–50% sparsity configuration achieves performance comparable to the 50%–50% baseline, whereas 50%–30% performs slightly worse. These findings suggest that prefill tokens can tolerate a lower sparsity ratio, while generation tokens benefit from a higher sparsity ratio.
>
> |Prefill ratio-Generation ratio|HumanEval $\uparrow$|
> |-|-|
> |10%-50%|39.63|
> |30%-50%|**40.85**|
> |50%-10%|35.98|
> |50%-30%|39.63|
> |50%-50%|**40.85**|

---

### Official Review · Reviewer_tBmW · 2025-10-28

**Soundness:** 2
**Presentation:** 3
**Contribution:** 2
**Rating:** 4
**Confidence:** 4

**Summary:**

This paper introduces a sparse attention mechanism designed to accelerate Diffusion Language Model (DLM) inference, particularly in long-context, high-step settings. While it is the first or one of the first approaches to apply sparse attention for DLMs, it is based on mainly *empirical* observations and shows only incremental improvements.

**Strengths:**

- Empirical Validation of DLM Dynamics: The analysis highlighting the temporal consistency and the importance of early denoising steps is highly insightful and provides foundation for a DLM-specific acceleration technique. The ablation study confirming the necessity of the "Skipping Sparse" component is well-executed and validates the central hypothesis regarding diffusion step sensitivity.

- State-of-the-Art Accuracy: the method achieves an almost lossless performance, competitive with other SOTA methods.

**Weaknesses:**

- *(General)* The paper is based mainly on very empirical observations concerning the distribution of attention scores. There is no validation of these properties through a quantitative analysis.
- *(General)* The method per-se is not very novel, as it applies principles that are well known already in ARMs (top-k selection based on attention scores in blocks) to DLMs.
- *(General)* The observation that attention patterns are head-specific is not a new empirical finding in the literature, and was already noticed in ARMs (cf. "Duo Attention", ICLR 2025 for KV Cache compression ARMs).
- *(Results)* The comparison against StreamingLLM looks ill-posed. Streaming LLM was developed for ARMs and is based on attention sinks in ARMs, but other work shows sinks in DLMs behave very differently (e.g. "Attention Sinks in Diffusion Language Models" Arxiv Preprint) so it is not surprising that it underperforms greatly in this scenario.
- *(Results)* The comparison with other baselines does not look very clear. Sparse methods are strongly dependent on the sparsity threshold, but the main results are displayed with a fixed chosen threshold for each method, making it difficult to understand the effective trade-off between accuracy and sparsity. A proper comparison should include different sparsity levels, which imply different FLOPs and memory footprint tradeoffs.
- *(Results)* Accuracy and Latency are evaluated against different sets of baselines which makes the results part a bit confusing.

**Questions:**

-  I think the choice of the baselines is not clear. This is a sparse attention method but the chosen baselines for efficiency include KVCaching methods for DLMs. However, KVCaching methods mainly aim at storing pre-computed values, while sparse attention aims at reducing computations. I think the paper would benefit from a clearer discussion of these two different aspects. Are they complementary, orthogonal ? How do they relate ?
-  For the same reason, it is not clear to me if this method is reducing memory footprint other than improving latency ? In line 294 the authors claim that the method reduced memory footprint but not experiments or comparisons are shown in this direction.

---

> ### Author Response · Authors · 2025-11-22
> **Author Response [1/2]**
>
> `>>>Q1` **Quantitative analysis for Attention Pattern.**
>
> `>>>A1` Thanks for the great suggestion! To strengthen our argument regarding attention patterns, we provide additional quantitative analysis:
>
> - **Attention Simlarity across steps**. To understand why attention scores remain similar over time, we analyze the Q and K matrices in `Appendix A.2`. The results show that this phenomenon primarily stems from the inherent similarity in Q and K.
> - **Consistent sparse patterns**. To verify whether precomputed sparse patterns remain stable during reuse, we conduct a study in `Section 4.3 (line 449–466)` using Jaccard similarity. The results indicate that the patterns maintain **over 90% similarity**, confirming their consistency throughout diffusion steps.
>
> Overall, these results are largely expected because, in DLM, **each step modifies only a few input tokens**. With such a small change, the internal representations—and therefore the attention—stay very similar from one step to the next.
>
>
> `>>>Q2` **Novelty.**
>
> `>>>A2` The reviewer is correct that certain techniques, such as top‑k selection, are not entirely new. However, the core novelty of our work lies in its *new problem formulation*, *new empirical findings*, and the *new method for DLMs*
>
> - **New problem setup**: To our knowledge, this is the **first work that studies sparse attention in DLMs** and leverages it to accelerate them.
> - **New Observation**: We uncover three **previously unreported properties** of attention patterns unique to DLMs.
> - **New method**: Building on these observations, our method (SparseD) applies full attention in the early steps, pre‑computes dynamic sparse patterns once, and then reuses them throughout later steps. This reuse of pre‑computed sparse attention patterns is, to our knowledge, **entirely new** in the context of DLMs.
>
> In summary, while top‑k selection is indeed a familiar tool, in our work it plays only a supporting role. The main contribution is the complete sparse‑attention framework tailored for DLMs. This goes much beyond simply transplanting ARM techniques to DLMs.
>
>
> `>>>Q3` **Why head-specific pattern is novel observation.**
>
> `>>>A3` Thank you for the comment.  While prior work, such as `Duo Attention`, has reported head-specific attention patterns in ARMs, our study is the **first** to systematically identify and analyze such patterns in DLMs.
>
> This distinction is **important**: DLMs differ substantially from ARMs in decoding behavior, and thus their head-wise attention characteristics cannot be assumed to transfer automatically. Our findings reveal new and previously unexplored properties of DLM attention, providing useful insights for better understanding and exploiting head-specific behaviors in future DLM research. We cite `Duo Attention` you mentioned in `Section 3 (line 189)`.
>
>
>
> `>>>Q4` **Comparison with StreamingLLM**
>
> `>>>A4` Thank you for the question! Although StreamingLLM is expected to underperform on DLMs, it is still a reasonable baseline for two key reasons:
> - **No prior work in DLMs**: since **no prior work** has investigated sparse attention specifically for DLMs, we adopt StreamingLLM—the most widely used sparse-attention method in AR models—as a representative baseline.
> - **Supporting our finding**: this comparison directly supports one of our important findings: sparse attention patterns in DLMs **differ significantly** from those in AR models. The limitations of StreamingLLM on DLMs underscore the necessity of developing DLM-specific sparse strategies such as ours.
>
> **Note**: We appreciate the reviewer pointing out the paper `Attention Sinks in Diffusion Language Models`, which provides a detailed and comprehensive analysis of attention sinks in DLMs. While we value its contribution, it was released **after our submission**, so our work provides earlier experimental evidence highlighting the incompatibility of attention sinks in DLMs. We have cited and discussed this paper in our revision (`lines 374–375`).
>
>
>
> `>>>Q5` **Sparsity Threshold.**
>
> `>>>A5` Thank you for the question. The reviewer is correct that the sparsity threshold is important in comparing sparse attention methods fairly.
>
> In practice, we have ensured that each sparse attention method is configured to be comparable in terms of the sparsity ratio. For instance, we set the sliding-window size to 2048 for the 4k evaluation and 4096 for the 8k evaluation so that their effective sparsity ratio is comparable to that of SparseD, ensuring a fair comparison. The full configurations for all sparse attention baselines are summarized in `Table 4`.

---

> ### Author Response · Authors · 2025-11-22
> **Author Response [2/2]**
>
> `>>>Q6` **Baseline for Accuracy and Latency.**
>
> `>>>A6` Thank you for the insightful comment. It is true that we use different baseline sets for evaluation. Specifically,
>
> - Accuracy is evaluated using both *sparse attention methods* and *efficient DLM methods*.
> - Latency, however, is evaluated **only within the family of sparse attention methods**.
>
> The reason we do not compare latency across different families of efficient DLM methods is that they **assume fundamentally different workloads**, making such comparisons unfair.
>
> **For example**: KV caching avoids recomputing past keys and values, reducing both key/value generation and attention computation. In contrast, sparse attention still performs full attention but reduces cost through sparsity. These two execution modes create very different workloads, so their latency is not meaningfully comparable.
>
>
>
> `>>>Q7` **Selection of Baselines.**
>
> `>>>A7` Thanks for pointing this out! We acknowledge that comparisons should ideally remain within the same family of sparse-attention baselines. However, to our knowledge, **no sparse attention** methods have been applied to DLMs before. Therefore, we included other relevant methods in our comparison:
>
> - **Choice of Baselines:** To provide meaningful baselines, we therefore include: **Sparse-attention methods** from ARMs (e.g., StreamingLLM), which are the closest existing techniques; **Efficient-DLM methods**, which mainly rely on KV caching. Including both types ensures a complete evaluation of our method in terms of both sparse-attention design and practical DLM performance.
> - **Relation between two types of methods**: Sparse attention and KV caching are orthogonal. In experiments, our method can indeed be combined with the cache method (details in `Appendix A.5`). This is because caching stores pre-computed keys/values to avoid redundant work, while sparse attention reduces the amount of attention computation itself. Therefore, they are different types of methods and orthogonal.
>
>
>
> `>>>Q9` **Memory Saving vs Latency Improvement.**
>
> `>>>A9` The reviewer is correct that our method primarily reduces memory usage rather than latency during attention computation. This benefit comes from our block-wise selection mechanism, which processes the attention map in small blocks instead of the full attention map at once.
>
> To support this argument, we added a **new memory usage analysis** in `Section 4.3 (Table 3)`. The results show that our method achieves memory consumption comparable to `FlashAttention`, and **significantly lower** than computing the full attention map without block-wise selection.

---

### Official Review · Reviewer_C5KP · 2025-10-30

**Soundness:** 4
**Presentation:** 4
**Contribution:** 3
**Rating:** 6
**Confidence:** 3

**Summary:**

This paper proposes SparseD, a sparse-attention framework specifically designed for diffusion language models (DLMs).
The authors first present three empirical observations:
1. Attention patterns are head-specific rather than uniform across heads.
2. Within each head, the attention pattern is stable across denoising steps.
3. Early denoising steps are critical, and applying sparsity too early degrades quality.

Based on these insights, SparseD adopts a simple and practical design:
- Skip sparsity during the early diffusion steps.
- At a representative step, compute full attention once, select block-wise Top-ρ attention regions per head, and store these as reusable sparse masks.
- Reuse the same mask across later steps, with isolated selection for prefill and generation tokens to better handle different distributions.

Experiments on Dream-7B-Instruct and LLaDA-1.5 show that SparseD maintains almost identical accuracy  while achieving significant speedup over FlashAttention at 64 k context and up to 1024 diffusion steps.
Ablation studies confirm that skipping early steps and reusing fixed patterns are both necessary for preserving model quality.

Overall, SparseD offers a simple yet effective mechanism to accelerate DLM inference while retaining quality, contributing a valuable empirical and practical perspective to efficient diffusion-based text generation.

**Strengths:**

- Novel DLM-specific insights: Identifies and validates three key empirical properties (head-specific, step-stable, early-step-sensitive) that distinguish DLM attention from AR models.
- Simplicity and practicality: The “skip-early + reuse mask + isolated selection” design is lightweight, easily implementable on top of Flash/FlexAttention, and hardware-friendly.
- Strong empirical results: Demonstrates consistent latency improvements with negligible performance drop across multiple DLMs and tasks (MMLU, GSM8K, HumanEval, RULER).
- Comprehensive ablations: Validates each component’s necessity and explores sensitivity to skip ratio and sparsity rate ρ.
- Clear presentation: Includes concise pseudo-code and diagrams explaining mask generation and reuse; writing is crisp and reproducible.

**Weaknesses:**

1. Overhead accounting clarity.  The paper amortizes sparse-pattern pre-computation but does not quantify the actual wall-clock fraction of that cost across steps.

2. Storage footprint not quantified.  The memory cost for storing per-head, per-layer sparse masks (after block-wise Top-ρ selection) is not reported, leaving uncertainty about scalability under 64 k contexts and many layers.

3. Comparison to dynamic sparsity baselines.  The paper argues autoregressive (AR) sparsity patterns do not transfer to diffusion models, but does not include an adapted dynamic head-wise Top-ρ baseline for DLMs to confirm the advantage of static reuse.

**Questions:**

1. What is the per-layer/head memory footprint of the stored sparse masks under your standard setting?
2. Have you tried computing masks at multiple checkpoints (e.g., twice in the diffusion process) and reusing them thereafter?
3. If the prompt distribution shifts (e.g., code → narrative), do the pre-computed masks still perform well?
4. How do latency and accuracy trade off with smaller or larger block sizes for a fixed ρ?
5. Can heads with low cross-step similarity be selectively excluded from mask reuse to improve robustness?

---

> ### Author Response · Authors · 2025-11-22
> **Author Response**
>
> `>>>Q1` **Time of Precomputation.**
>
> `>>>A1` Thank you for your suggestion. Pre-computation accounts for roughly 6.3%–13.7% of the total time in SparseD across different sequence lengths. Although there is some extra cost, it is far smaller than the latency reduction and can be further amortized as the number of diffusion steps increases. The detailed report of precomputing time is in `Section 4.3 (line 469–499)`.
>
>
>
> `>>>Q2` **Memory usage report.**
>
> `>>>A2` Thank the reviewer for your brilliant suggestion! As suggested, we report the storage associated with sparse attention patterns in `Section 4.3 (line 469~499)`. The additional storage accounts for **only 0.3%** in 64k context length.
>
>
>
> `>>>Q3` **Comparison to dynamic sparsity baselines.**
>
> `>>>A3` Thanks for the suggestion! We already provide this comparison in `Table 2`, which shows that static reuse outperforms dynamic sparsity in terms of latency.
>
> Specifically, to verify this, we compare SparseD with reuse (Static) and without reuse (Dynamic). As shown in `Table 2 (line 2 vs. 4)`, both achieve strong accuracy; However, removing reuse introduces substantial latency because **sparse patterns must be recomputed at every step**. This demonstrates that static reuse is highly effective for reducing latency while preserving accuracy.
>
>
> `>>>Q4` **Per-layer/head memory footprint.**
>
> `>>>A4` We truly thank you for the question. Please see `A2`.
>
>
>
> `>>>Q5` **Computing masks at multiple steps.**
>
> `>>>A5` Thank you for the question.
> Our method does **not necessitate** computing masks at multiple checkpoints (diffusion steps) because the mask patterns exhibit stability across steps.
>
> As suggested, we added an experiment where sparse patterns are pre-computed at $T \times \text{skip}\$% and $T \times 2 \times \text{skip}\$% steps and then reused thereafter. The results, shown below, indicate that computing the patterns twice achieves similar performance as computing them once. This outcome validates that a single computation of masks is sufficient to achieve excellent performance.
> |LLaDA-1.5|RULER-4k|
> |-|-|
> |Twice|90.80|
> |SparseD (Once)|90.86|
>
>
>
> `>>>Q6` **Prompt distribution shifts.**
>
> `>>>A6` Thank you for your insightful question. Our method can achieve great performance in different prompt distribution. This is beacause our method computes sparse patterns for each prompt **during inference**, rather than pre-computing them before inference. Hence, it is not affected by prompt distribution.
>
> We evaluate on three dataset types: general language understanding (MMLU), mathematical reasoning (GSM8K), and code generation (HumanEval). As shown in `Table 1`, our method’s accuracy remains consistent across datasets, demonstrating that our method effectively handle diverse prompts.
>
>
>
> `>>>Q7` **Trade off in block sizes.**
>
> `>>>A7` Thank you for the question. We added an experiment to evaluate latency and accuracy with different block sizes at a fixed $\rho$, as shown below.
>
> All three block size configurations achieve similar accuracy. However, smaller blocks lead to significantly higher latency, while larger blocks offer a better trade-off between latency and accuracy. This is because
> the block size directly governs the efficiency of the GPU's memory hierarchy and computational kernel launch overhead. Hence, based on these results, we choose 128 as our block size.
> |Block Size|RULER-4k|Latency (64k)|
> |-|-|-|
> |32|89.41|1915|
> |128|90.86|1695|
> |256|90.03|1661|
>
>
> `>>>Q8` **Excluding low similar mask.**
>
> `>>>A8` Great insight! In theory, enabling full attention in attention heads with low cross-step similarity could indeed improve robustness. However, because all attention heads within a layer are **executed in parallel**, adopting full attention in certain attention heads would cause the latency to revert to that of full attention—thereby eliminating the efficiency benefits of sparsity. Therefore, although feasible, this strategy would significantly degrade overall inference speed.

---

### Official Review · Reviewer_ibCB · 2025-11-02

**Soundness:** 3
**Presentation:** 3
**Contribution:** 2
**Rating:** 4
**Confidence:** 3

**Summary:**

This paper proposes **SparseD**, a **training-free, inference-time** sparse attention method for **Diffusion Language Models (DLMs)**. The key observations are: (1) **head-specific** attention patterns, (2) **high step-wise consistency** within a head, and (3) **early steps are sensitive** to sparsification. SparseD thus applies **full attention in early steps**, then **pre-computes head-specific sparse patterns once** using **block-wise top-ρ% selection with isolated selection for prefill vs. generation tokens**, and **reuses** the pattern thereafter. Experiments on Dream-7B-Instruct and LLaDA-1.5 show **up to 1.50× speedup** over FlashAttention at 64k context and 1,024 steps with negligible accuracy loss.

**Strengths:**

**Practical & training-free:** No retraining; one-shot sparse mask + reuse; early-step full attention preserves quality.
- **Simple, hardware-friendly design:** Block-wise avg-pool → top-ρ% selection; **isolated selection** for prefill/generation.
- **Consistent wins at scale:** Clear latency gains at long contexts; speedups increase with more steps due to reuse.

**Weaknesses:**

## Major
- **Theoretical under-pinning is light.** No bounds/guarantees on error from mask reuse or head-specific sparsity; argument is largely empirical.
- **Model diversity is limited.** Only two DLMs (Dream-7B, LLaDA-1.5); generality to other DLM families or hybrid architectures is unproven.
- **Early-step sensitivity.** Skip ratio is tuned (20–30%) rather than principled; transferability across schedules/steps remains unclear.

## Minor
- **Amortized precompute cost** is not fully quantified in wall-clock breakdowns, though reuse intuition is sound.
- **Context vs. cache baselines.** A deeper head-to-head with cache-based methods across sequence lengths would help.
- **AR-sparse comparisons.** Positioning vs. structured AR patterns (e.g., Longformer/BigBird) could be elaborated.

**Questions:**

1. **Skip ratio sensitivity:** How does accuracy/speed trade-off vary beyond 20–30% and across different total steps (e.g., 256 vs. 2048)? Any adaptive criterion?
2. **Dynamic prompts/streaming:** If prefill/generation boundaries change mid-run, can the mask be updated without losing reuse benefits?
3. **Precompute overhead:** What is the exact one-time cost for mask construction at 64k and how does it scale with heads/layers? (A timing table would help.)
4. **Composability with caches:** Any synergy or redundancy when combining SparseD with dKV-Cache/Fast-dLLM for short vs. long contexts?
5. **Very long contexts (128k+):** Does block granularity dominate selection quality; how does isolated selection behave there?

---

> ### Author Response · Authors · 2025-11-22
> **Author Response [1/2]**
>
> `>>>Q1` **Theoretical under-pinning is light.**
>
> `>>>A1` Thanks for the question! Although our paper contains little theoretical analysis, we believe those empirical evidence is sufficiently strong to support these arguments:
>
> - **Sparse reuse**: `Figure 1` shows that attention patterns exhibit high cosine similarity across steps. To further verify whether the pre-computed sparse pattern in SparseD remains consistent during reuse, we add a **new quantitative analysis** in `Section 4.3 (line 450–466)`. The results show that similarity stays **above 90%**, confirming that the pattern is highly stable and that sparse reuse is effective.
> - **Head-specific sparsity**: While our ratio is empirical, it is based on performance analysis across different sparsity levels (`Figure 7(b, c)`). These results show the selected ratio **(30%) provides the best trade-off** between accuracy and latency.
>
>
>
> `>>>Q2` **Model diversity is limited.**
>
> `>>>A2` Thanks for the suggestion! At submission time, we chose the strongest DLMs available (Dream-7B, LLaDA-1.5), but we definitely agree that broader evaluation is important.
>
> As suggested, we tested our method on additional models, including the multimodal DLM, **LLaDA-V**. As shown in the table below, we set $\rho = 0.3$ and $skip = 20$%. The results demonstrate that our method achieves lossless performance on the `mmmu_val` and `chartqa` benchmarks. This again confirms its effectiveness beyond text-only DLMs.
>
> Of course, if there are other specific models the reviewer recommend, we would greatly appreciate your input. Thanks again!
>
> **Note**: Our method is not designed for hybrid architectures (e.g., DLM+AR), so we did not include those.
>
> |LLaDA-V|mmmu_val (T=2) $\uparrow$|chartqa (T=16) $\uparrow$|
> |-|-|-|
> |Origin|48.67|78.32|
> |SparseD|48.56|78.12|
>
>
>
> `>>>Q3` **Early-step sensitivity.**
>
> `>>>A3` Thanks for the suggestion! As requested, we conducted an ablation study on the `ruler_qa_squad` dataset to test different skip ratios across various steps. The results (shown below) indicate a consistent trend: larger skip ratios generally yield better performance.
>
> In the paper, we selected a moderate range of **20–30%**, which appears transferable across different step counts.
> |ruler_qa_squad|skip=10% $\uparrow$|20% $\uparrow$|30% $\uparrow$|50% $\uparrow$|
> |-|-|-|-|-|
> |32 Steps|82.40|83.40|83.20|83.77|
> |64 Steps|86.17|87.30|86.77|87.47|
> |128 Steps|81.07|81.33|81.43|81.63|
>
>
>
> `>>>Q4` **Amortized precompute cost.**
>
> `>>>A4` Good question! The precompute cost is relatively small; It accounts for **roughly 6.3%–13.7% of the total time** in SparseD across different sequence lengths. While there is some overhead, it is much smaller than the latency reduction achieved and can be further amortized as the number of diffusion steps increases. Detailed analysis is in `Section 4.3 (line 469–499)`.
>
>
>
> `>>>Q5` **Context vs. cache baselines.**
>
> `>>>A5` Thank you for the insightful suggestion. As suggested, we added **new comparison** with cache-based methods across sequence lengths in `Appendix A.5`. For a fair comparison, we combined SparseD with a cache method and compared it against dKV-Cache and Fast-dLLM.
>
> The results show that integrating SparseD with caching can **further accelerate inference**, but leads to a drop in performance. Full details are provided in `Appendix A.5`. In future work, we will explore ways to better integrate sparse attention and cache-based methods to achieve lossless acceleration.
>
>
>
> `>>>Q6` **AR-sparse comparisons.**
>
> `>>>A6` We sincerely appreciate the reviewer’s suggestion. We added a comparison with Longformer and BigBird in terms of accuracy on RULER-4k, shown below. Both models perform worse than ours.
>
> This comparison highlights the limitations of AR-based sparse attention in DLMs while demonstrating the strong performance of SparseD. Since Longformer and BigBird are well-established sparse-attention methods in ARs, discussing them in DLMs will provide valuable insights. We have cited and discussed them in `Section 2 (lines 113–115)`.
>
> |Block Size|RULER-4k$\uparrow$|
> |-|-|
> |Longformer|75.91|
> |Big Bird|60.04|
> |SparseD|**90.89**|
>
>
>
> `>>>Q7` **Skip ratio sensitivity.**
>
> `>>>A7` We truly thank you for the question. Please see `A3`.
>
>
>
> `>>>Q8` **Dynamic prompts/streaming.**
>
> `>>>A8` We appreciate the reviewer for the new question. In fact, our method is originally **designed for typical fixed-length DLMs** and does not currently handle boundary-changing settings. These fixed-length models remain the most common, including popular ones like Dream and LLaDA.
>
> While we acknowledge that some models allow dynamic boundaries, this is beyond the scope of our current work. We plan to explore this direction in future research.

---

> ### Author Response · Authors · 2025-11-22
> **Author Response [2/2]**
>
> `>>>Q9` **Precompute overhead.**
>
> `>>>A9` We sincerely appreciate the suggestions. Across different sequence lengths, pre-computation cost is relatively small and accounts for **roughly 6.3%–13.7%** of SparseD’s total runtime (**note that** current DLMs remain limited in scale, so we report how this overhead scales with context length). While this introduces some additional cost, it is far outweighed by the overall latency reduction and can be further amortized over multiple diffusion steps. A detailed analysis is provided in `Section 4.3 (lines 469–499)`.
>
>
>
> `>>>Q10` **Composability with caches.**
>
> `>>>A10` Thank you for the insightful question. Our method can indeed be combined with KV caching to **achieve further acceleration**. However, this integration introduces a noticeable performance drop. Detailed experimental results and analysis are provided in `Appendix A.5`.
>
> **Why does this happen?**
>
> - Sparse attention and KV caching operate at different levels of the model and are largely orthogonal.
> - **Positive side**: They can be combined easily.
> - **Negative side**: Both techniques introduce some degree of approximation or information loss. When combined, **these errors can accumulate**, leading to a more significant performance degradation.
>
> In future work, we plan to investigate strategies to better integrate these approaches and achieve lossless acceleration.
>
>
>
> `>>>Q11` **Block granularity.**
>
> `>>>A11` Thank you for the question. Block granularity *does not significantly impact performance*, and isolated selection proves effective across all granularities.
>
> In the table below, we evaluate the effect of block granularity on selection quality and isolated selection. Since DLMs like Dream and LLaDA underperform with very long contexts, we use RULER with a 4k context for this evaluation. The results show that while block granularity slightly influences accuracy, all configurations benefit from isolated selection, highlighting its effectiveness.
> |Block Size|w/ isolated selection $\uparrow$|w/o isolated selection $\uparrow$|
> |-|-|-|
> |32|89.41|89.33|
> |128|90.86|90.53|
> |256|90.03|89.76|

---

> > ### Comment · Reviewer_ibCB · 2025-11-27
> >
> > Dear Authors,
> >
> > Thank you for your detailed and kind response. My concerns are mostly addressed, and I will raise my rating to 6.
> >
> > I have one minor request:
> > Could you update the LLaDA-V results both with and without Sparse-D, and ensure that all revised results are consistently reflected in the updated manuscript to improve completeness?

---

> > > ### Author Response · Authors · 2025-11-27
> > > **Response to Reviewer ibCB**
> > >
> > > Dear Reviewer ibCB,
> > >
> > > Thank you sincerely for your positive feedback and support. We have now included a detailed evaluation in the updated manuscript (specifically in `Appendix A.6`). This result demonstrates the effectiveness of our approach in the advanced multimodal DLM, LLaDA-V.
> > >
> > > Your constructive guidance has been invaluable to our work. Thank you once again for your guidance.
> > >
> > > If you have any further questions or concerns, we would be more than pleased to continue the discussion at any time.
> > >
> > > Sincerely,
> > >
> > > The Authors

---

### Author Response · Authors · 2025-11-22
**Official Comment by Authors**

We sincerely thank all reviewers for their constructive feedback. We deeply appreciate the following positive comments:

- Novel insights: `C5KP`, `tBmW`, `Q4L3`
- Simple and practical method: `ibCB`, `C5KP`, `Q4L3`
- Strong empirical results: `ibCB`, `C5KP`, `tBmW`, `Q4L3`
- Good ablation study: `C5KP`, `tBmW`, `Q4L3`
- Clear presentation: `C5KP`

We will address the specific questions and concerns raised by the reviewers in the subsequent sections of this rebuttal.

---

### Author Response · Authors · 2025-12-01
**Summary of Discussions with Reviewers Prior to the Leakage Incident**

Dear Area Chair,

Thank you for your time and effort in managing the review process for our paper. We understand that this requires significant additional effort, and we are truly grateful for your time and support. To facilitate your review, we would like to provide a brief summary of the discussion period.

---
- **In the pre-discussion state**, the reviewers provided positive feedback on several key aspects:
    - Novel insights: `C5KP`, `tBmW`, `Q4L3`
    - Simple and practical method: `ibCB`, `C5KP`, `Q4L3`
    - Strong empirical results: `ibCB`, `C5KP`, `tBmW`, `Q4L3`
    - Good ablation study: `C5KP`, `tBmW`, `Q4L3`
    - Clear presentation: `C5KP`

---
- **During the discussion phase**, we actively addressed all constructive questions and suggestions raised by the reviewers. We are pleased to report that **`Reviewer ibCB`** was fully satisfied with our rebuttal and **increased the rating from 4 to 6 \!**
    - **Rating Change: 4 $\rightarrow$ 6 \!**
    - **Please note**: This rating increase occurred **prior to the wide OpenReview anonymity breach**.

---
- **Response to common questions/suggestion:**
    - **Lack resource report of pre-compution stage (`ibCB`, `C5KP`, `tBmW`, `Q4L3`)**: In `Section 4.3 (line 469–499)`, we added report of latency, memory usage during pre-compution to show the **high efficiency** of our approach in pre-compution stage.
    - **More quantitive analysis of attention patterns (`tBmW`, `Q4L3`)**: To strengthen our argument regarding attention patterns, we provided **additional** quantitative analysis: **Firstly**, we analyze the Q and K matrices in `Appendix A.2` to show that high similarity of attention patterns primarily stems from the inherent similarity in Q and K. **Secondly,** we conduct a **new study** in `Section 4.3 (line 449–466)` using Jaccard similarity to verify whether precomputed sparse patterns remain stable during reuse. The results indicate that the patterns maintain **over 90% similarity**, confirming their consistency throughout diffusion steps.

    In addition to these common points, we have addressed all specific concerns raised by each reviewer and posted detailed responses in the comment sections. Meanwhile, **we have incorporated the necessary changes into the revised manuscript**. All modifications are highlighted in blue for ease of reference.

---
Finally, we sincerely thank you for your extra effort in managing the recent OpenReview emergency. We deeply appreciate your dedication to ensuring a fair review process during this challenging time.

Sincerely,

The Authors

---

### Meta-Review · Area_Chair_y6FN · 2026-01-06

**Summary:**

## Strengths
* The proposed method is training-free and easy to implement, which makes it practical for deployment.
* The method is evaluated across multiple benchmarks (MMLU, GSM8K, HumanEval, RULER).
* Ablation studies are thorough and clearly presented.
* The paper is clearly written and easy to follow.
* Demonstrates efficiency gains in long-context settings, which increase with the number of denoising steps.


## Concerns
1. Insights and method design
	1. The method skips the early steps, and the skip ratio is tuned rather than principled; transferability across schedules/steps remains unclear [Reviewer ibCB].
	2. The empirical observation on the attention pattern is not surprising. Similar observations were noticed in the ARM or DM's sparse attention literature [Reviewer tBmW].
	3. The method is not novel, as it applies similar techniques with prior methods in ARM and DMs [Reviewer tBmW].
	4. The pre-computation step might be inefficient for truly massive context (e.g., 256k, 1M) due to its quadratic complexity [Reviewer Q4L3].
	5. If prefill/generation boundaries change mid-run, can the mask be updated without losing reuse benefits? [Reviewer ibCB]
2. Analysis
	1. Lack deeper analysis and theoretical exploration beyond empirical observations [Reviewer ibCB,tBmW,Q4L3].
	2. Overhead should be better analyzed [Reviewer ibCB,C5KP].
3. Results
	1. Should experiment with more DLM families and cache-based baselines [Reviewer ibCB].
	2. Should compare to dynamic sparsity baseline [Reviewer C5KP].
	3. The comparison against StreamingLLM is not well-aligned, as StreamingLLM is designed for ARMs. The reviewer mentioned one concurrent work shows sinks in DLMs behave differently [Reviewer tBmW].
	4. Accuracy and Latency are evaluated against different sets of baselines, making the results confusing [Reviewer tBmW].

**Reviewer Concerns:**

For concerns on insights and model design:
* Concern 1.1 is **addressed** by the author rebuttal that provides experiment with different #timesteps showing the 20%-30% choice works well.
* Concern 1.2 is **not well addressed**. The three key observations in this paper are indeed similar to existing work in autoregressive language model and image diffusion model's acceleration literature. Therefore, they are not surprising nor interesting. Nevertheless, whether these insights are novel is a matter of opinion.
* Concern 1.3 is **not well addressed**. The authors argue they are the first work that studies sparse attention in DLMs and leverages it to accelerate them. This is true. However, the early skipping technique, similar techniques of pre-computing attention pattern or other feature and then reusing in the later timesteps in diffusion models acceleration literature. The novelty is limited.
* Concern 1.4 is **not well addressed**. The authors mention Table 3 in the rebuttal revision. However the table doesn't contain experiments corresponding to reviewer's suggested 256k/1M settings.
* Concern 1.5 is **answered** to be out of this single paper's scope. I agree it's OK to only discuss this.

For concerns on the analysis:
* Concern 2.1 is **partially addressed**. The authors add analysis on Q/K similarity to go one-step further to explain the empirical observation on the attention similarity. But still, the analysis and theoretical insights is shallow and cannot enlighten obvious future exploration.
* Concern 2.2 is **addressed**.

For concerns on results:
* Concern 3.1 is **addressed** by the author rebuttal that provides new experiment on LLaDA-V.
* Concern 3.2 is **addressed**.
* Concern 3.3 is **answered** by the author response that the reviewer's mentioned work is after the paper submission and is concurrent. And the authors add discussion to the related work in the revision.
* Concern 3.4 is **answered**.

**Reviewer Scores:**

* Reviewer ibCB initially gave 4, and **will likely increase the score to 6**.
* Reviewer tBmW initially gave 4, and I guess the reviewer **will not increase the score**, as the rebuttal doesn't straightly address the novelty issue (e.g., instead of directly saying DLM is a totally different domain, the authors should directly relate and discuss those very similar insights and techniques in ARM and DM acceleration).
* Reviewer Q4L3 and C5KP initially gave 6, and I think they **will keep the original positive score**.

Overall, this paper receives a borderline score. As the authors address most of the concerns in the author response, after discussion with SAC, we recommend acceptance.

---

### Decision · Program_Chairs · 2026-01-26

Accept (Poster)